# Calibrated Physics-Informed Uncertainty Quantification

**Vignesh Gopakumar** [1 2]  **Ander Gray** [3]  **Lorenzo Zanisi** [2]  **Timothy Nunn** [2]  **Daniel Giles** [1]  **Matt J. Kusner** [4 5]
**Stanislas Pamela** [2]  **Marc Peter Deisenroth** [1]

## Abstract

Simulating complex physical systems is crucial for understanding and predicting phenomena across diverse fields, such as fluid dynamics and heat transfer, as well as plasma physics and structural mechanics. Traditional approaches rely on solving partial differential equations (PDEs) using numerical methods, which are computationally expensive and often prohibitively slow for real-time applications or large-scale simulations. Neural PDEs have emerged as efficient alternatives to these costly numerical solvers, offering significant computational speed-ups. However, their lack of robust uncertainty quantification (UQ) limits deployment in critical applications. We introduce a model-agnostic, physics-informed conformal prediction (CP) framework that provides guaranteed uncertainty estimates without requiring labelled data. By utilising a physics-based approach, we can quantify and calibrate the model's inconsistencies with the physics rather than the uncertainty arising from the data. Our approach utilises convolutional layers as finite-difference stencils and leverages physics residual errors as nonconformity scores, enabling data-free UQ with marginal and joint coverage guarantees across prediction domains for a range of complex PDEs. We further validate the efficacy of our method on neural PDE models for plasma modelling and shot design in fusion reactors.

## 1. Introduction

Numerical PDE solvers are essential tools in scientific and engineering simulations (Danabasoglu et al., 2020; Giudi-

[1]Centre for Artificial Intelligence, University College London [2]Computing Division, UK Atomic Energy Authority [3]Heudiasyc Laboratory [4]Polytechnique Montréal [5]Mila - Quebec AI Institute. Correspondence to: Vignesh Gopakumar <v.gopakumar@ucl.ac.uk>.

*Proceedings of the 42^{nd} International Conference on Machine Learning*, Vancouver, Canada. PMLR 267, 2025. Copyright 2025 by the author(s).

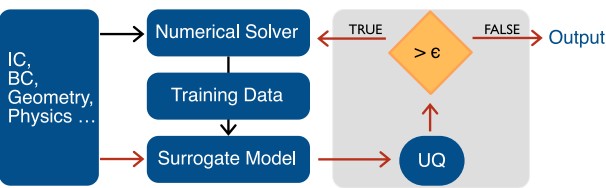

*Figure 1.* **Neural PDE framework:** Neural PDE solvers use data from traditional numerical solvers to quickly approximate PDEs across various conditions (shown by black arrows). To ensure reliability, these models incorporate uncertainty quantification (UQ) methods. If the predicted error exceeds a coverage threshold $\epsilon$, the numerical solver is utilised, further adding to the training data; otherwise, predictions are used as output (shown by red arrows). This paper focuses on developing a new UQ method for assessing confidence in neural PDE models, emphasising the grey-shaded region of the framework.

celli et al., 2024), but their computational demands and environmental impact pose significant challenges (Horwitz, 2024). Machine learning approaches have emerged as efficient alternatives for modelling/emulating PDEs (Bertone et al., 2019; Karniadakis et al., 2021), successfully deployed across weather forecasting (Kochkov et al., 2024; Meyer et al., 2022; Giles et al., 2024), fluid dynamics (Jiang et al., 2020; Pfaff et al., 2021), and nuclear fusion applications (Poels et al., 2023; Carey et al., 2024; Gopakumar & Samaddar, 2020). Neural PDE solvers provide rapid approximations, but present a critical cost-accuracy trade-off. While generating outputs consistently, their solutions may violate physical constraints or produce misleading results with high confidence (Gopakumar et al., 2023a). A typical neural-PDE framework (see Figure 1) trains surrogate models on numerical simulation data to predict the evolution of PDE under various conditions, with uncertainty quantification (UQ) methods reverting to numerical solvers when predictions fail coverage thresholds. However, current UQ methods lack statistical guarantees (Zou et al., 2024), require extensive simulation data (Gopakumar et al., 2024a), or require architectural modifications (Abdar et al., 2021).

To address these limitations, we propose a framework combining PDE residuals over neural PDEs with Conformal Prediction (CP) to provide uncertainty estimates that guarantee coverage. Our approach evaluates Physics Residual Errors

(PRE) from neural PDE solver predictions and performs calibration using marginal and joint CP formulations. Our method provides statistically valid coverage within the residual space (Vovk et al., 2005), offering error bounds against the violation of physical conservation laws. The framework is model-agnostic (applicable to all types of surrogate models) and does not require additional data generated from expensive numerical simulations. It yields interpretable uncertainty bounds indicating the model's physics-guided inconsistencies, addressing the (over)confidence issue of neural PDE solvers (Zou et al., 2024). Our contributions:

**Calibrated physics-informed UQ:** A novel physics-informed nonconformity metric using PDE residuals. This quantifies uncertainty through physical inconsistency rather than training data variation, providing input-independent prediction sets while relaxing exchangeability restrictions.

**Marginal and Joint CP:** Our approach guarantees coverage bounds both marginally (univariate per dimension) and jointly (multivariate across the entire prediction domain), enabling the identification of volatile predictions and creating a rejection-acceptance criteria.

## 2. Related Work

Recently, CP, as a method of performing UQ, has been gaining popularity for usage with spatio-temporal data (Sun, 2022). Several works have explored the inductive CP framework for spatial and sequential data (Stankeviciute et al., 2021; Xu & Xie, 2021; Xu et al., 2023), including in the operator space (Ma et al., 2024). Gopakumar et al. (2024a) extends the marginal-CP framework to pre-trained as well as to fine-tuned surrogate models for physical system modelling across an infinite-dimensional setting. Alternatively, error bounds for PDE surrogates have been devised by Gray et al. (2025) using set propagation to project the singular value decomposition of the prediction error onto the prediction space.

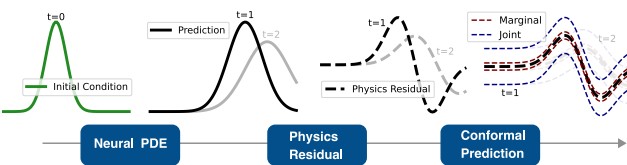

*Figure 2.* Schematic of physics-informed uncertainty quantification workflow. Initial conditions generate neural PDE predictions autoregressively, over which physics residual errors are estimated. Calibration via marginal and joint conformal prediction yields error bars - pointwise for marginal-CP and domain-wide for joint-CP.

The usage of PDE residuals under the guise of Physics-Informed Machine Learning (PIML) (Karniadakis et al., 2021) was made popular as an optimisation strategy for Physics-Informed Neural Networks (PINNs) (Raissi et al.,

2019) and has found application in optimising neural operators (Li et al., 2024) and soft/hard enforcement of the physical constraints to deep learning models (Du et al., 2024; Chalapathi et al., 2024). However, they have rarely been used as a tool for providing UQ to surrogate models, and where they have found application, UQ remained uncalibrated (Zhu et al., 2019). The majority of the literature in UQ for neural PDE solvers has been looking at Bayesian methods, such as dropout, Bayesian neural networks, and Monte Carlo methods (Geneva & Zabaras, 2020; Zou et al., 2024; Psaros et al., 2023), which lack guarantees or are computationally expensive.

## 3. Background

### 3.1. Neural PDE Solvers

Consider the generic formulation of a PDE modelling the spatio-temporal evolution of $n$ field variables $u \in \mathbb{R}^n$ across a range of initial conditions:

$$D = D_t(u) + D_X(u) = 0, \quad X \in \Omega, \ t \in [0, T], \quad (1)$$
$$u(X, t) = g, \quad X \in \partial\Omega, \quad (2)$$
$$u(X, 0) = a(\lambda, X). \quad (3)$$

Here, $X$ defines the spatial domain bounded by $\Omega$, $[0, T]$ the temporal domain, and $D_X$ and $D_t$ the composite operators of the associated spatial and temporal derivatives. The PDE is further defined by the boundary condition $g$ and initial condition $a$, which can be parameterised by $\lambda$. The set of solutions of field variables is expressed as $u \in \mathcal{U}$.

Neural PDE solvers as surrogate models learn the behaviour governed by Equation (1) using a parameterised neural network $\mathcal{N}\mathcal{N}_\theta$. Starting from the initial conditions, the network is trained to solve the spatio-temporal evolution of the fields given by $\Omega \cup [0, T]$. Neural operators $\mathcal{N}\mathcal{O}_\theta$ are a special class of neural networks that learn the operator mapping from the function space of the PDE initial conditions $a \in \mathcal{A}$ to the function space of solutions $u \in \mathcal{U}$. A neural operator for solving an initial-value problem can be expressed as

$$\mathcal{U} = \mathcal{N}\mathcal{O}_\theta(\mathcal{A}),$$
$$u(X, t) = \mathcal{N}\mathcal{O}_\theta\Big(u(X, 0), t\Big),$$
$$X \in \Omega, \ t \in [0, T]. \quad (4)$$

A **Fourier Neural Operator** (FNO) is an autoregressive neural operator that learns the spatio-temporal evolution of PDE solutions by leveraging the Fourier transform as the kernel integrator (Li et al., 2021). The field evolution is learned using tunable weight matrices of the network, parameterised directly in the Fourier space of the PDE solutions.

Since CP and our extension of it provide a post-hoc measure of quantifying the uncertainty of a neural PDE, it remains

agnostic to model choice and training conditions. Considering the model independence of our approach, we restrict our experiments to modelling PDEs with an FNO. The FNO is chosen due to its cost-accuracy trade-off and efficiency as demonstrated by de Hoop et al. (2022) and Gopakumar et al. (2023b). CP over a range of neural-PDE solvers has been applied by Gopakumar et al. (2024a), who also demonstrate that the coverage guarantees are upheld irrespective of the model choice, further necessitating us to not experiment with various model architectures.

### 3.2. Conformal Prediction

Conformal prediction (CP) (Vovk et al., 2005; Shafer & Vovk, 2008) is a statistical framework that addresses the accuracy of a predictive model. Consider a machine learning model $\hat{f} : \mathcal{X} \to \mathcal{Y}$ trained on a dataset $(X_i, Y_i)_{i=1}^{N}$, that can be used to predict the next true label $Y_{n+1}$ at query point $X_{n+1}$. CP extends the point prediction $\mathcal{P} : \tilde{Y}_{n+1}$ to a prediction set $\mathbb{C}^{\alpha}$, ensuring that

$$\mathbb{P}(Y_{n+1} \in \mathbb{C}^{\alpha}) \geq 1 - \alpha. \qquad (5)$$

This coverage guarantee, a function of the user-defined confidence level $\alpha$, holds irrespective of the chosen model and training dataset. The only condition is that the calibration samples and the prediction samples are exchangeable. Traditional inductive CP partitions the labelled data into training and calibration sets (Papadopoulos, 2008). The performance of the model on the latter, measured using a *nonconformity score*, is used to calibrate the model and obtain prediction sets.

Conventionally, nonconformity scores act on the model predictions and a labelled dataset (Kato et al., 2023). For deterministic models, they are often formulated as the Absolute Error Residual (AER) of the model predictions $\hat{f}(X)$ and targets $Y$. For probabilistic models, the score function (STD) is the absolute value of the z-score (with the prediction mean, standard deviation and target given as $\hat{f}_{\mu}(X), \hat{f}_{\sigma}(X), Y$ respectively. Having obtained a distribution of nonconformity scores $\hat{s}$ of the calibration dataset $(X_i, Y_i)_{i=1}^{n}$, a quantile $\hat{q}$ corresponding to the desired coverage $1 - \alpha$ is estimated from its cumulative distribution function $F_{\hat{s}}$ (Papadopoulos, 2008):

$$\hat{q^{\alpha}} = F_{\hat{s}}^{-1}\left( \frac{\lceil (n+1)(1-\alpha) \rceil}{n} \right). \qquad (6)$$

The quantile estimates the error bar associated with the desired coverage and is combined with the new prediction to obtain the prediction sets. The nonconformity score functions and their prediction sets for AER and STD are given in Table 1. Both AER and STD are data-intensive CP methods, requiring calibration data that converges to a beta distribution (Angelopoulos & Bates, 2023). This data dependence

*Table 1.* Overview of nonconformity metrics AER, STD, and PRE and their corresponding score functions and prediction sets.

| | Score Function ($\hat{s}$) | Prediction Sets ($\mathbb{C}^{\alpha}$) |
|---|---|---|
| AER | $\left( \lvert \hat{f}(X_i) - Y_i \rvert \right)_{i=1}^{n}$ | $\hat{f}(X_{n+1}) \pm \hat{q}^{\alpha}$ |
| STD | $\left( \frac{\lvert \hat{f}_{\mu}(X_i) - Y_i \rvert}{f_{\sigma}(X_i)} \right)_{i=1}^{n}$ | $\hat{f}_{\mu}(X_{n+1}) \pm \hat{q}^{\alpha} \, \hat{f}_{\sigma}(X_{n+1})$ |
| PRE | $\left( \lvert D(\hat{f}(X_i)) \rvert \right)_{i=1}^{n}$ | $\pm \hat{q}^{\alpha}$ |

restricts their application to domains where sufficient data exists a priori or can be easily obtained

## 4. Physics Residual Error (PRE)

We introduce a novel *data-free nonconformity score* based directly on the PDE for surrogate models. The Physics Residual Error (PRE) is defined as the PDE residual (Saad & Schultz, 1986) estimated over the discretised PDE solution obtained from the surrogate model. For an abstract PDE as in Equation (1), the PDE residual is the evaluation of the composite differential operator $D$ over a field(s) $u$ under the influence of an external force $b$

$$\mathcal{D}(u) - b = 0, \qquad (7)$$

The PDE residual is treated as a score function by taking its L1 norm as indicated in Table 1. While well-defined PDEs have solutions obeying Equations (1) to (3), numerical solutions often fail to converge to the true solution (Pinder, 2018). Neural PDEs, trained on approximate numerical data, are further prone to non-convergent predictions. In numerical analysis, the norm of the PDE residual is often used as a criterion for stability, convergence, and accuracy (Iserles, 2009). The PRE typically represents the violation of conservation laws associated with the physical system. Using the residual error as a nonconformity score quantifies the neural PDE solver's non-convergence to the physical ground truth of the PDE. By further using conformal prediction, we obtain coverage bounds over this conservative, residual space with guaranteed coverage.

The norm $\lvert D(\mathcal{N}\mathcal{O}_{\theta}(u)) \rvert$ of the residual operator itself provides a measure of UQ for the neural PDE. However, it is limited by the accuracy of the gradient estimation method and can become computationally expensive when exploring a vast solution space (Tolsma & Barton, 1998). By using the residual norm as a nonconformity score, we further calibrate the approximate physics residual error that an inexpensive and coarse differential operator obtains. CP using PRE provides statistically valid and guaranteed error

bars across the PDE's residual space, incorporating physical information into the calibration procedure and providing a calibrated measure of the physical misalignment of the surrogate model.

PRE as a nonconformity score enables *data-free conformal prediction*. The estimated scores rely only on the neural PDE predictions and not on the target as in AER and STD. The only criterion is that the calibration and prediction domains arise from exchangeable PDE conditions. As shown in Table 1, PRE gives "prediction sets" independent of the prediction inputs (see Appendix B for formalism). Traditional CP methods rely on calibration using labelled data to construct prediction intervals that contain the true values with a specified confidence level $\alpha$. These methods guarantee that the true solution will lie within the estimated error bounds based on this empirical calibration. Contrastingly, our CP-PRE formulation takes a fundamentally different approach. Instead of requiring target data for calibration, we leverage the unique property of PDEs where the true solution in the residual space exists at zero. This eliminates the need for empirical calibration data. Our method focuses on ensuring that predictions themselves fall within coverage bounds $\mathbb{C}^\alpha$, rather than guaranteeing that the true solution lies within these bounds. This allows us to validate prediction sets without access to ground truth data—a significant advantage over traditional CP approaches. We formalise this novel property in our theoretical framework presented in Appendix A.

### 4.1. Marginal-CP

The CP formulation was initially conceptualised for calibrating univariate functions with single-point outputs (Vovk et al., 2005). It has recently been extended to spatio-temporal data, with multi-dimensional outputs with an immutable tensor structure (Gopakumar et al., 2024a). Within such spatio-temporal settings, CP has been implemented to provide marginal coverage, i.e. the calibration procedure provides independent error bars for each cell within the spatio-temporal domain. For an output tensor $Y \in \mathbb{R}^{N_x \times N_y \times N_t}$, where $N_x, N_y, N_t$ represent the spatio-temporal discretisation of the domain, marginal-CP uses the non-conformity scores outlined in Table 1 across each cell of $Y$ to obtain error bars, which will be compliant with Equation (5) for each cell. Marginal-CP using PRE helps indicate spatio-temporal regions within individual predictions that lie outside the calibrated bounds of physics violation and require specific attention, treating those prediction regions with caution.

### 4.2. Joint-CP

The joint-CP formulation constructs a calibration procedure that provides coverage bands for multivariate functions.

These coverage bands expand across the entire simulation domain $\Omega \times [0, T]$ (discretised as $\mathbb{R}^{N_x \times N_y \times N_t}$) rather than an individual cell within it. For a coverage band $\mathbb{C}^\alpha$, the joint-CP formulation ensures that $1 - \alpha$ predictions/solutions lie within the bounds. For performing joint-CP, the non-conformity scores are modified to reflect the supremum of the score functions $\hat{s}$ in Table 1. They are modulated by the standard deviation $\sigma$ of the calibration scores (Diquigiovanni et al., 2021) to obtain prediction bands with varying widths based on local behaviour (Diquigiovanni et al., 2022). The modifications of the score functions and prediction sets to perform CP are given by

$$\mathcal{S}_{\text{joint}} = \sup_{X \in \Omega, \, t \in [0, T]} \left( \frac{\hat{s}}{\sigma(\hat{s})} \right), \tag{8}$$

$$\mathbb{C}^\alpha_{\text{joint}} = p(X_{n+1}) \pm \hat{q}^\alpha \cdot \sigma(\hat{s}), \tag{9}$$

where $\hat{s}$ and $p(X_{n+1})$ are the formulations of the nonconformity scores and the prediction at $X_{n+1}$ used for marginal-CP as shown in Table 1. Joint-CP becomes particularly useful in identifying predictions that fail to fall within coverage, allowing us to accept or reject a prediction based on a predetermined probability. Similar to that demonstrated by Casella et al. (2004), our framework can perform acceptance-rejection using a CP-based criterion. The acceptance probability is based on confidence level $\alpha$. If a prediction is rejected, the PDE parameters that led to those predictions could be provided to the expensive physics-based numerical PDE solver for further evaluation as indicated in Figure 1.

### 4.3. Differential Operator: Finite-Difference Stencils as Convolutional Kernels

Calibrating neural PDEs using PRE nonconformity scores requires frequent evaluations of the composite differential operator $D$ in Equation (1). For PDEs, this involves estimating spatio-temporal gradients across the discretised domain, ranging from millions in simple cases to billions of gradient operations for complex physics. To address this computational challenge, we developed a scalable gradient estimation method for evaluating PRE.

We use convolution operations with Finite Difference (FD) stencils as convolutional kernels for gradient estimation (Actor et al., 2020; Chen et al., 2024a;b). For instance, the 2D Laplacian operator $\nabla^2$, using a central difference scheme with discretisation $h$, can be approximated by

$$\nabla^2 \approx \frac{1}{h^2} \begin{bmatrix} 0 & 1 & 0 \\ 1 & -4 & 1 \\ 0 & 1 & 0 \end{bmatrix} \tag{10}$$

and used as a kernel. This approach is justified by the mathematical equivalence of FD approximations and discrete convolutions. Both represent matrix-vector multiplications of a block Toeplitz matrix with a field vector (Strang, 1986;

Fiorentino & Serra, 1991). The efficiency of this method stems from the optimised implementation of convolution operations in machine learning libraries like PyTorch (Paszke et al., 2019) and TensorFlow (Abadi et al., 2015). [1]

The FD approximation offers several advantages over Automatic Differentiation (AD) for our application. It is compatible with CP as a post-hoc measure, requires no architectural modifications, and is model-agnostic. Furthermore, FD implemented via convolutions is more memory-efficient than AD, which requires storing the entire computational graph. As we focus on the (mis)alignment of neural PDEs with Equation (1), we disregard boundary conditions, but demonstrate that they can be accounted for with convolutional padding in Appendix E (Alguacil et al., 2021). Utilising finite difference schemes, the residuals are estimated up to the truncation errors of the Taylor approximation (MacKinnon & Johnson, 1991). Although discretisation plays a role in the width of the error bars, CP-PRE still guarantees coverage, and this is further explored in Appendix D.1.

## 5. Experiments

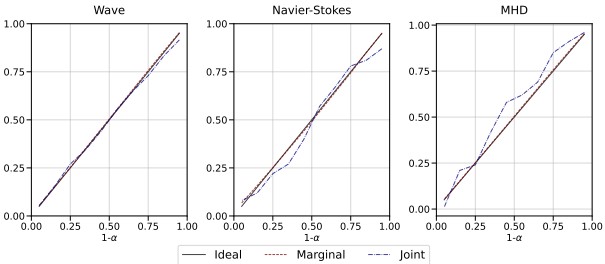

*Figure 3.* **Validation plots demonstrating coverage guarantee** detailed in Equation (5) obtained by performing CP using PRE across experiments. The average empirical coverage obtained experimentally is given on the $y$-axis (ranging from 0 to 1, with 1 representing 100% coverage), while the theoretical coverage is represented on the $x$-axis. We obtain guaranteed coverage while using marginal-CP formulation and near-to-ideal coverage for the joint-CP formulation.

CP-PRE experiments comprise two campaigns. First, we benchmark CP-PRE within standard neural PDEs (Section 5.1 to 5.3). The calibration process (Figure 2) involves: (a) sampling inputs to the model to generate predictions, (b) calculating PRE(s) scores, and (c) calibrating physical error using marginal and joint-CP formulations. Validation uses the same PDE condition bounds as calibration. Within this campaign, we compare our method (CP-PRE) with other methods of providing uncertainty estimation for neural-

PDEs. We compare various Bayesian methods along with multivariate inductive conformal prediction to our method as shown in Table 2. We demonstrate that our method is capable of providing valid coverage guarantees for both in and out-of-distribution testing without a significant increase in evaluation times (including time taken for sampling in Bayesian methods, data generation in data-driven CP and subsequent calibration). Methods that attain the required coverage are emboldened. Table 2 provides a qualitative comparison of our method against the other benchmarks and highlights that our method is data-free, requires no modification or sampling and provides guaranteed coverage in a physics-informed manner. We confine our comparison studies in Table 3, 4, and 5 to the Wave, Navier-Stokes and Magnetohydrodynamic equations.

The second campaign (Section 5.4, 5.5) applies CP-PRE to fusion applications. We enhance tokamak plasma behaviour surrogate models to identify erroneous dispersion regions (Section 5.4) and integrate CP-PRE with tokamak design surrogates to identify viable designs and areas needing additional simulations (Section 5.5). This campaign demonstrates the utility of CP-PRE in complex, practical applications. One-dimensional PDE experiments demonstrating CP-PRE are demonstrated in Appendix F and can be reproduced using code in the supplementary material.

### 5.1. Wave Equation

The two-dimensional wave equation is given by

$$\frac{\partial^2 u}{\partial t^2} = c^2 \left( \frac{\partial^2 u}{\partial x^2} + \frac{\partial^2 u}{\partial y^2} \right). \tag{11}$$

We solve the field $u$ given in Equation (11) within the domain $x, y \in [-1, 1]$, $t \in [0, 1.0]$ with wave velocity $c = 1.0$ using a spectral solver with periodic boundary conditions (Canuto et al., 2007). The initial conditions are parameterised by the amplitude and position of a Gaussian field. A 2D FNO is trained on this data to predict 20-time steps autoregressively from a given initial state. Figure 4 compares the model predictions against ground truth, showing the PRE and confidence bounds from marginal and joint-CP at 90% coverage. The PRE reveals noise artefacts in regions where the field should be zero, highlighting physical inconsistencies despite apparently accurate predictions. Joint-CP bounds are necessarily larger than marginal-CP bounds as they span across the spatio-temporal domain as opposed to being cell-wise. As demonstrated in Figure 3, both CP approaches achieve the expected coverage guarantees. Table 3 highlights the superior performance of CP-PRE over other methods of UQ, where it guarantees coverage when evaluated for in and out-of-distribution without requiring any additional data, drastically reducing the evaluation time. Marginal-CP shows linear coverage due to cell-wise averag-

---

[1] The Basic Linear Algebra Subroutines (BLAS) in these libraries leverage vectorisation and efficient memory access for significant performance gains. Our experiments demonstrate a 1000x speedup using *torch.nn.functional.conv3d* versus an equivalent *numpy* finite difference implementation on standard CPU.

| Method | Data-Free | Modification-Free | Sampling-Free | Guaranteed Coverage | Physics-Informed |
|---|---|---|---|---|---|
| MC Dropout (Gal & Ghahramani, 2016) | ✓ | ✗ | ✗ | ✗ | ✗ |
| Deep Ensemble (Lakshminarayanan et al., 2017) | ✓ | ✗ | ✗ | ✗ | ✗ |
| BNN (MacKay, 1992) | ✓ | ✗ | ✗ | ✗ | ✗ |
| SWA-G (Maddox et al., 2019) | ✓ | ✗ | ✗ | ✗ | ✗ |
| CP-AER (Gopakumar et al., 2024a) | ✗ | ✓ | ✓ | ✓ | ✗ |
| CP-PRE (Ours) | ✓ | ✓ | ✓ | ✓ | ✓ |

*Table 2.* Comparing features across various UQ measures. Our method is data-free, does not require any modifications or sampling, and helps obtain guaranteed coverage bounds in a physics-informed manner.

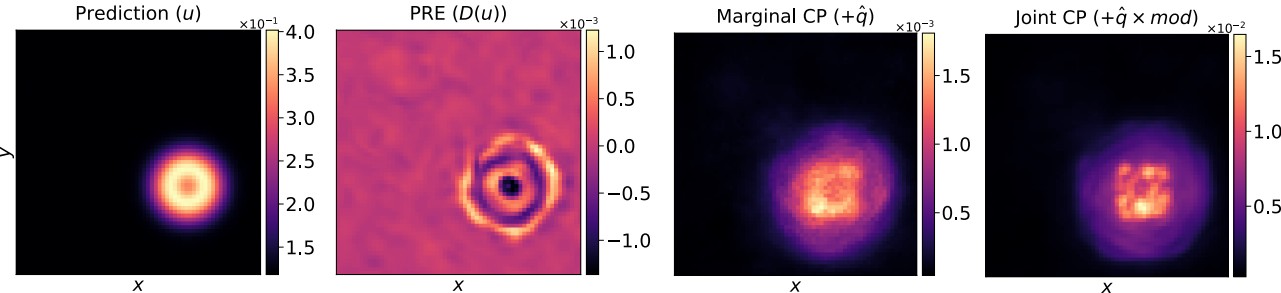

*Figure 4.* **Wave:** (From left to right) neural PDE (FNO) prediction at the last time instance, physics residual error of the prediction, Upper error bars obtained by performing marginal-CP and joint-CP respectively (90% coverage). For brevity, we have only shown the upper error bars of the symmetric prediction sets. *mod* represents the modulation function in Table 1. The physical inconsistencies within the residual space of the prediction are calibrated and bounded using CP-PRE.

ing. At the same time, joint-CP exhibits coverage variations that depend on the calibration dataset (see Table 3 for stability analysis across multiple calibration sets). Additional experimental details are provided in Appendix J.

### 5.2. Navier-Stokes Equation

Consider the two-dimensional Navier-Stokes equations

$$\vec{\nabla} \cdot \vec{v} = 0, \tag{12}$$

(Continuity equation)

$$\frac{\partial \vec{v}}{\partial t} + (\vec{v} \cdot \vec{\nabla})\vec{v} = \nu\nabla^2\vec{v} - \nabla P, \tag{13}$$

(Momentum equation)

where we are interested in modelling the evolution of the velocity vector ($\vec{v} = [u, v]$) and pressure ($P$) field of an incompressible fluid with kinematic viscosity ($\nu$). For data generation, Equations (12) and (13) are solved on a domain $x \in [0, 1]$, $y \in [0, 1]$, $t \in [0, 0.5]$ using a spectral-based solver (Canuto et al., 2007). A 2D multi-variable FNO (Gopakumar et al., 2024b) is trained to model the evolution of velocity and pressure autoregressively up until the $20^{th}$ time instance.

Unlike the previous example, the Navier-Stokes case has two equations with corresponding PRE estimates: continuity (Equation (12)) and momentum (Equation (13)) for mass and momentum conservation. Our CP-PRE method calibrates model deviation from physical ground truth for

each equation. Table 4 shows that Bayesian methods fail to provide coverage for FNO modelling, while multivariate CP (CP-AER) provides coverage but requires substantially higher evaluation time arising from the generation of simulation data for calibration. Figures 20 and 21 show PRE bounds from both equations. Having two PDE residuals enables easier rejection of predictions violating both bounds. Physics details, parameterisation, and training are in Appendix K. Within the scope of this paper, we limit ourselves to measuring the deviation of the model with the PDE residual. The CP-PRE formulation can be extended to obtain bounds for both the initial and boundary conditions, further explored within Appendix E.

### 5.3. Magnetohydrodynamics

Consider the magnetohydrodynamic (MHD) equations

$$\frac{\partial \rho}{\partial t} + \vec{\nabla} \cdot (\rho\vec{v}) = 0, \quad \text{(Continuity equation)} \tag{14}$$

$$\rho\left(\frac{\partial \vec{v}}{\partial t} + \vec{v} \cdot \nabla\vec{v}\right) = \frac{1}{\mu_0}\vec{B} \times (\vec{\nabla} \times \vec{B}) - \nabla P, \tag{15}$$

(Momentum equation)

$$\frac{\mathrm{d}}{\mathrm{d}t}\left(\frac{P}{\rho^\gamma}\right) = 0, \quad \text{(Energy equation)} \tag{16}$$

$$\frac{\partial \vec{B}}{\partial t} = \vec{\nabla} \times (\vec{v} \times \vec{B}), \quad \text{(Induction equation)} \tag{17}$$

$$\vec{\nabla} \cdot \vec{B} = 0, \quad \text{(Gauß law for magnetism)} \tag{18}$$

| UQ | in-distribution | | out-of-distribution | | Time | |
| --- | --- | --- | --- | --- | --- | --- |
| | MSE | Coverage (95%) | MSE | Coverage (95%) | Train (hr) | Eval (s) |
| Deterministic | 1.77e-05 ± 3.69e-07 | - | 2.46e-03 ± 2.00e-05 | - | 0:38 | 22 |
| MC Dropout | 1.44e-04 ± 3.26e-06 | **97.31 ± 0.03** | 2.12e-03 ± 2.60e-05 | 89.83 ± 0.07 | 0:52 | 120 |
| Deep Ensemble | 8.76e-06 ± 2.43e-07 | **98.02 ± 0.04** | 2.42e-03 ± 1.58e-05 | 83.44 ± 0.12 | 3:10 | 112 |
| BNN | 1.92e-04 ± 1.92e-06 | **97.10 ± 0.09** | 2.67e-03 ± 1.26e-05 | 91.76 ± 0.10 | 0:53 | 118 |
| SWA-G | 1.41e-05 ± 1.74e-06 | **94.55 ± 3.25** | 2.55e-03 ± 2.82e-05 | 81.90 ± 3.31 | 0:47 | 113 |
| CP-AER | 1.76e-05 ± 4.40e-07 | **95.70 ± 0.21** | 2.46e-03 ± 1.41e-05 | **95.59 ± 0.14** | 0:38 | 2022 |
| CP-PRE (Ours) | 1.78e-05 ± 4.61e-07 | **95.52 ± 0.21** | 2.46e-03 ± 1.25e-05 | **95.39 ± 0.12** | 0:38 | **32** |

*Table 3.* Wave Equation — CP-PRE guarantees coverage across distributions while providing the quickest evaluation time.

| UQ | in-distribution | | out-of-distribution | | Time | |
| --- | --- | --- | --- | --- | --- | --- |
| | MSE | Coverage (95%) | MSE | Coverage (95%) | Train (hr) | Eval (s) |
| Deterministic | 1.05e-04 ± 6.91e-06 | - | 3.67e-03 ± 5.30e-05 | - | 3:22 | 25 |
| MC Dropout | 5.96e-04 ± 2.30e-05 | 82.21 ± 0.22 | 4.30e-03 ± 8.05e-05 | 44.05 ± 0.26 | 3:34 | 153 |
| Deep Ensemble | 1.22e-04 ± 3.95e-06 | 91.31 ± 0.08 | 3.67e-03 ± 3.52e-05 | 30.74 ± 0.19 | 16:22 | 147 |
| BNN | 6.90e-03 ± 1.31e-04 | 89.91 ± 0.20 | 6.95e-03 ± 1.31e-04 | 85.19 ± 0.23 | 3:39 | 152 |
| SWA-G | 1.96e-04 ± 1.15e-05 | 84.22 ± 2.37 | 3.63e-03 ± 1.37e-04 | 31.00 ± 2.85 | 3:28 | 146 |
| CP-AER | 1.05e-04 ± 6.58e-06 | **95.56 ± 0.40** | 3.66e-03 ± 2.81e-05 | **95.54 ± 0.15** | 3:22 | 20026 |
| CP-PRE (Ours) | 1.07e-04 ± 5.18e-06 | **95.44 ± 0.22** | 3.70e-03 ± 4.23e-05 | **95.57 ± 0.14** | 3:22 | **134** |

*Table 4.* Navier-Stokes Equations — CP-PRE guarantees coverage across distributions while providing the quickest evaluation time.

where the density ($\rho$), velocity vector ($\vec{\mathbf{v}} = [u, v]$) and the pressure of plasma is modelled under a magnetic field ($\vec{\mathbf{B}} = [B_x, B_y]$) across a spatio-temporal domain $x, y \in [0,1]^2$, $t \in [0,5]$. $\mu_0$ is the magnetic permeability of free space. Equations (14) to (18) represent the ideal MHD equations as a combination of the Navier-Stokes equations for fluid flow with Maxwell's equations of electromagnetism (Alfvén, 1942; Gruber & Rappaz, 1985; Mocz et al., 2014). The equations assume perfect conductivity (no magnetic diffusivity) and no viscosity. We focus our experiment on the modelling of the Orszag-Tang vortex of a turbulent plasma (Orszag & Tang, 1979) with the data being generated using a finite volume method (Eymard et al., 2000). A 2D FNO is trained to model the evolution of all 6 variables over a dataset generated by parameterised initial conditions.

Equations (14) to (18) provide us with five measures of estimating the PRE of the MHD surrogate model. Each PRE estimate depends on a different set of variables associated with the system, allowing us to infer errors contributed to each variable accordingly. Table 5 demonstrates that the Bayesian methods fail miserably in providing valid error bars over the MHD case. The CP-AER provides guaranteed coverage but is heavily dependent on simulation data, adding to the computational expense. CP-PRE using induction (Equation (17)) and energy (Equation (16)) are shown for 90% coverage ($\alpha = 0.1$), sliced at $y = 0.5m$. The plots show PRE along the x-axis with marginal and joint bounds at a specific simulation time. Marginal bounds

are significantly tighter, while joint bounds are wider but useful for identifying predictions that violate conservation equations. Additional CP plots using other residuals (Figures 27 and 28) and physics/surrogate model details are in Appendix L.

### 5.4. Plasma Modelling within a Tokamak

In (Gopakumar et al., 2024b), the authors model the evolution of plasma blobs within a fusion reactor (known as a tokamak) using an FNO. They explore the case of electrostatic modelling of reduced magnetohydrodynamics with data obtained from the JOREK code (Hoelzl et al., 2021). In the absence of magnetic pressure to confine it, the plasma, driven by kinetic pressure, moves radially outward and collides with the wall of the reactor. The plasma is characterised by density $\rho$, electric potential $\phi$ and Temperature $T$, and the FNO models their spatio-temporal evolution autoregressively. Borrowing that pre-trained model and utilising the reduced-MHD equations within the toroidal domain, we demonstrate obtaining calibrated error bars using CP-PRE at scale. The FNO demonstrated in (Gopakumar et al., 2024b) can model the plasma six orders of magnitude faster than traditional numerical solvers, and by providing calibrated error bars over the predictions, a wider range of plasma configurations can be validated.

We focus on the temperature equation within reduced-MHD (equation 3 within (Gopakumar et al., 2024b)) as it comprises all the variables associated with the plasma. As shown

| | in-distribution | | out-of-distribution | | Time | |
|---|---|---|---|---|---|---|
| UQ | MSE | Coverage (95%) | MSE | Coverage (95%) | Train (hr) | Eval (s) |
| Deterministic | 2.20e-03 ± 5.20e-03 | - | 4.71e-02 ± 1.06e-03 | - | 5:00 | 40 |
| MC Dropout | 3.29e-02 ± 5.86e-04 | 41.13 ± 0.19 | 2.09e-01 ± 1.38e-03 | 16.91 ± 0.06 | 5:30 | 240 |
| Deep Ensemble | 3.59e-03 ± 3.51e-04 | 78.15 ± 0.16 | 3.41e-01 ± 3.15e-02 | 39.63 ± 0.31 | 26:25 | **235** |
| BNN | 4.20e-03 ± 4.08e-05 | 90.24 ± 0.10 | 4.63e-02 ± 8.98e-04 | 62.37 ± 0.46 | 5:40 | 240 |
| SWA-G | 2.61e-03 ± 9.68e-05 | 48.50 ± 3.81 | 4.53e-02 ± 6.64e-04 | 14.22 ± 1.35 | 5:22 | 236 |
| CP-AER | 2.20e-03 ± 4.38e-05 | **95.61 ± 0.26** | 4.69e-02 ± 8.18e-04 | **95.60 ± 0.27** | 5:00 | 40042 |
| CP-PRE (Ours) | 2.20e-03 ± 4.96e-03 | **95.54 ± 0.18** | 4.71e-02 ± 1.06e-03 | **95.67 ± 0.22** | 5:00 | 482 |

*Table 5.* Magnetohydrodynamic Equations — CP-PRE guarantees coverage across distributions with only a marginal increase in evaluation time, arising from residual evaluation over a larger family of PDEs (see Equations (14) to (18)).

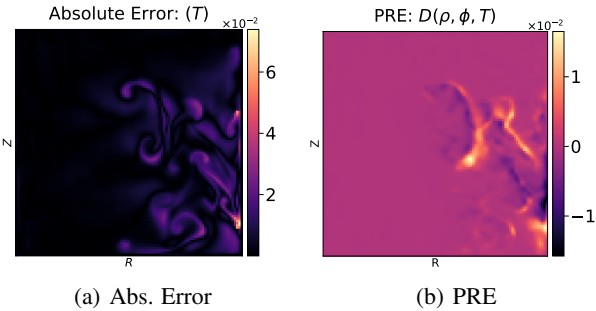

(a) Abs. Error          (b) PRE

*Figure 5.* **Reduced MHD:** CP-PRE using the Temperature equation (Eqn. 3 in (Gopakumar et al., 2024b)) of reduced-MHD to bound the plasma surrogate models. The PRE captures the model error relatively well, allowing us to provide lower and upper error bars corresponding to our required coverage.

in Figure 5, our method can capture the model error across a range of predictions and devise error bars that provide guaranteed coverage without additional data. In figure 5(a), we demonstrate the absolute error in the model prediction of the temperature evolution, correlating that with the PRE over the temperature equations in figure 5(b). By obtaining bounds on the PRE, we can determine the efficacy of the surrogate model in evaluating plasma evolution and identify conditions under which it fails, running the MHD code JOREK under those failed conditions.

### 5.5. Magnetic Equilibrium in a Tokamak

Tokamaks confine plasma within a toroidal vessel using magnetic fields to achieve nuclear fusion. The plasma at high temperatures is contained by magnetic fields that counterbalance its kinetic pressure. This equilibrium state, a function of magnetic coil configurations and plasma parameters, is governed by the Grad-Shafranov (GS) equation (Somov, 2012)

$$\frac{\partial^2 \psi}{\partial r^2} - \frac{1}{r}\frac{\partial \psi}{\partial r} + \frac{\partial^2 \psi}{\partial z^2} = -\mu_0 r^2 \frac{dp}{d\psi} - \frac{1}{2}\frac{dF^2}{d\psi}. \quad (19)$$

where $\psi$ represents the poloidal magnetic flux, $p$ the kinetic pressure, $F = rB$ the toroidal magnetic field, and $\mu_0$ the magnetic permeability given in a 2D toroidal coordinate system characterised by $r$ and $z$. While traditional numerical solvers like EFIT++ and FreeGSNKE (Lao et al., 1985; Amorisco et al., 2024) are used for equilibrium reconstruction, their computational cost has motivated neural network alternatives (Joung et al., 2023; Jang et al., 2024). However, these surrogate models lack UQ capabilities, making their deployment within the control room challenging.

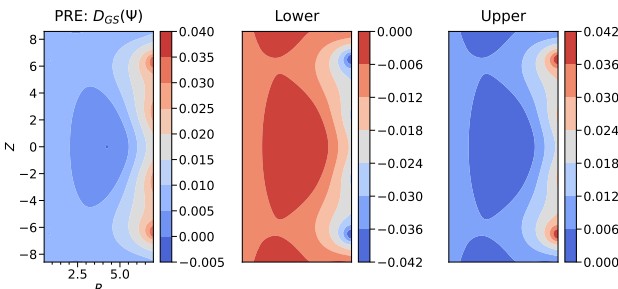

*Figure 6.* **Grad-Shafranov:** The PRE for a specific poloidal field coil configuration is indicated on the left, and the lower and upper bars for 50% are displayed adjacent to it. Aside from guaranteeing coverage, the CP-PRE framework allows us to discard physically inconsistent equilibria predicted by the surrogate model.

We implement an auto-encoder that maps poloidal magnetic flux across the poloidal cross-section for given tokamak architectures, conditioned on poloidal field coil locations under constant plasma current. While this accelerates simulation by 4 orders of magnitude, it lacks physical guarantees. By incorporating Equation (19) within the CP-PRE framework, we identify physically stable equilibria and obtain statistically valid error bounds. Figure 6 shows the PRE over a surrogate model prediction with lower and upper error bars for 50% coverage. Further details about the problem setting and the model can be found in Appendix N.

## 6. Extensions of CP-PRE beyond PDEs

Our framework applies to any prediction case where the forward problem can be formulated as a residual with an equality constraint (Appendices A and B). The framework works in any scenario where the forward model can be expressed in the standard canonical form

$$Ax - b = 0, \tag{20}$$

where $x$ is the model prediction, $A$ is a differential or algebraic operator governing the dynamics, and $b$ is a non-homogeneous term such as a function or a constant.

This extends our applications beyond PDEs to ODEs and algebraic equations found in control problems (Jiang & Jiang, 2014), chemical reactions (Thöni et al., 2025), biological systems (Wang et al., 2019), and financial scenarios (Liu et al., 2019).

We deliberately focus on PDEs as they represent the most comprehensive and challenging case—multi-dimensional domains with complex spatio-temporal dependencies and unique computational challenges. Success with PDEs implicitly validates applicability to simpler systems.

## 7. Discussion

If "All models are wrong, but some are useful" (Box, 1976), through this work, we explore a novel framework for providing a measure of *usefulness* of neural PDEs. We deploy a principled method of evaluating the accuracy of the solution, i.e. its (calibrated) obedience to the known physics of the system under study. As opposed to other methods of UQ for neural PDEs, our method has the unique advantage of being physics-informed. This allows us to study the physical inconsistencies of the model predictions with coverage guarantees provided by conformal prediction. We conclude with a discussion of the strengths, limitations and potential improvements.

**Strengths**   PRE estimates the violation of conservation laws in neural PDE predictions, guaranteed error bounds over the physics deviation. This post-hoc uncertainty quantification is model- and physics-agnostic, scaling linearly with model complexity and quasi-linearly with PDE complexity due to the additive nature of differential operators. Our framework reformulates CP to be data-free, expressing model inaccuracy solely through PRE, not requiring a labelled dataset. This approach reduces calibration costs and loosens exchangeability restrictions as we can modify the calibration and, hence, the prediction domain by simply reformulating the PRE accordingly. The PRE formulation (Section 4, Appendix B) yields input-independent prediction sets, allowing for the identification of weak prediction regions within single simulations (marginal-CP) and

across multiple predictions (joint-CP). The latter enables a rejection criterion for a set of predictions, potentially serving as an active-learning pipeline for neural PDE solvers (Musekamp et al., 2025). CP-PRE provides guaranteed coverage irrespective of the model, chosen discretisation, or the PDE of interest; however, the width of the error bar indicates quantitative features in the model quality. A well-trained model will exhibit tighter error bars as opposed to a poorer fit model; see Appendix I.

**Limitations**   Our method's coverage bounds exist in the PDE residual space rather than the Euclidean space of physical variables. Transforming to physical space involves challenging set propagation through integral operations, which may require approximations (Teng et al., 2023) or expensive Monte Carlo sampling (Andrieu et al., 2003). The data-free approach lacks a grounding target for calibration, though we argue that a large sample of model outputs provides a statistically significant overview of uncertainty. The sampling cost from the neural-PDE solver for calibration involves intensive gradient evaluations. PRE estimation using finite-difference stencils also introduces the errors associated with Taylor expansion. The current formulation is limited to regular grids with fixed spacing, though extensions to unstructured grids via graph convolutions are possible (Eliasof & Treister, 2020). CP-PRE does not help differentiate between aleatoric and epistemic uncertainty. It aligns with conformal prediction's characterisation of predictive uncertainty. From one perspective, this could be viewed as aleatoric uncertainty since we construct confidence intervals relative to a specific probability distribution (the distribution from which calibration data, i.e. initial conditions/PDE coefficients, are sampled). Alternatively, it could be considered epistemic uncertainty since we model the neural network's error through confidence intervals (typically used for unknown but fixed quantities). While we believe the latter interpretation is more appropriate, we acknowledge that the traditional aleatoric/epistemic dichotomy may not be directly applicable to our framework. This distinction is most valuable when both uncertainty types coexist and require separate treatment (Ferson & Ginzburg, 1996).

## 8. Conclusion

We address the problem of the reliability of neural-PDE solvers by proposing CP-PRE, a novel conformal prediction framework. Our method provides guaranteed and physics-informed uncertainty estimates for each cell within a prediction, identifying erroneous regions while discerning physically inconsistent predictions across the entire spatio-temporal domain. Our work enhances the reliability of neural PDE solvers, potentially broadening their applicability in science and engineering domains where robust uncertainty quantification is crucial.

## Impact Statement

This paper presents work whose goal is to advance the field of Machine Learning. There are many potential societal consequences of our work, none of which we feel must be specifically highlighted here.

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

## A. Theorem: Data-Free CP

**Preliminaries:** Let $D : \mathbb{R}^m \to \mathbb{R}^m$ be a physics residual operator mapping a function to its PDE residual value, where: $\{X_i\}_{i=1}^n$ is the calibration set, $\hat{f}$ is the model, $\hat{q}^\alpha$ is estimated as the $\lceil (n + 1)(1 - \alpha) \rceil / n$ -quantile of $\{|D(\hat{f}(X_i))|\}_{i=1}^n$

**Theorem A.1.** *If the residuals $\{D(\hat{f}(X_i))\}_{i=1}^{n+1}$ are exchangeable random variables, then for any significance level $\alpha \in (0, 1)$ and any new input $X_{n+1}$ we have the following coverage guarantee:*

$$\mathbb{P}(|\mathcal{D}(\hat{f}(X_{n+1}))| \in C_\alpha) \geq 1 - \alpha; \ \ C_\alpha = [-\hat{q}_\alpha, \hat{q}_\alpha]$$

*Proof.* Let $R_i = |D(\hat{f}(X_i))|$ for $i = 1, \ldots, n + 1$. We have, by assumption, $(R_1, \ldots, R_n, R_{n+1})$ is an exchangeable sequence. Define the rank $\pi$ of $R_{n+1}$ w.r.t. all other residuals:

$$\pi(R_{n+1}) = |\{i = 1, \ldots, n + 1 : R_i \leq R_{n+1}\}|$$

By exchangeability, the rank $\pi(R_{n+1})$ is uniformly distributed over $\{1, \ldots, n + 1\}$. Therefore,

$$P(\pi(R_{n+1}) \leq \lceil (n+1)(1-\alpha) \rceil) = \frac{\lceil (n + 1)(1 - \alpha) \rceil}{n} \geq 1 - \alpha.$$

By construction of $\hat{q}^\alpha$ we have that,

$$\{\pi(R_{n+1}) \leq \lceil (n + 1)(1 - \alpha) \rceil\} \subseteq \{R_{n+1} \leq \hat{q}^\alpha\}.$$

Putting this together,

$$P(|D(\hat{f}(X_{n+1}))| \leq \hat{q}^\alpha) = P(R_{n+1} \leq \hat{q}^\alpha) \geq 1 - \alpha,$$

which completes the proof. □

## B. PRE: Score Function and Prediction Sets

For a general nonconformity score $S$, the prediction set for a new input $X_{n+1}$ is typically defined as:

$$\mathbb{C}^\alpha(X_{n+1}) = \{y : S(X_{n+1}, y) \leq \hat{q}^\alpha\},$$

where $\hat{q}^\alpha$ is the $(1-\alpha)$-quantile of the nonconformity scores on the calibration set.

For AER and STD, the nonconformity scores depend on both the input $X$ and the output (target) $Y$:

$$S_{AER}(X, Y) = |\hat{f}(X) - Y|, \tag{21}$$

$$S_{STD}(X, Y) = \frac{|\hat{f}_\mu(X) - Y|}{\hat{f}_\sigma(X)}. \tag{22}$$

The resulting prediction sets are:

$$\mathbb{C}^\alpha_{AER}(X_{n+1}) = [\hat{f}(X_{n+1}) - \hat{q}^\alpha, \hat{f}(X_{n+1}) + \hat{q}^\alpha], \tag{23}$$

$$\mathbb{C}^\alpha_{STD}(X_{n+1}) = [\hat{f}_\mu(X_{n+1}) - \hat{q}^\alpha \hat{f}_\sigma(X_{n+1}),$$
$$\hat{f}_\mu(X_{n+1}) + \hat{q}^\alpha \hat{f}_\sigma(X_{n+1})]. \tag{24}$$

These prediction sets clearly depend on the input $X_{n+1}$.

For PRE, the nonconformity score depends only on the model output and not on the target:

$$S_{PRE}(\hat{f}(X)) = |D(\hat{f}(X)) - 0|,$$

where $D$ is the PDE residual operator. The key difference is that the true output $Y$ for PRE, irrespective of the PDE is always 0 and does not depend on the input $X$. PRE is a measure of how well the model output satisfies the physics rather than how it fits certain data. Hence, we can formulate a nonconformity score that is data-free and eventually leads to input-independent prediction sets as given below.

For PRE, we can reframe the prediction set definition:

$$\mathbb{C}^\alpha_{PRE} = \{\hat{f}(X) : |D(\hat{f}(X))| \leq \hat{q}^\alpha\}.$$

This set is not defined in terms of the true $Y$ values but in terms of the allowable model outputs $\hat{f}(X)$ that satisfy the PDE residual constraint. Thus, the prediction set can be expressed as:

$$\mathbb{C}^\alpha_{PRE} = [-\hat{q}^\alpha, \hat{q}^\alpha].$$

This formulation is independent of the input $X$, as it only depends on the quantile $\hat{q}^\alpha$ derived from the calibration set as given in Equation (6).

To validate predictions using PRE:

1. For a new input $X_{n+1}$, compute $\hat{f}(X_{n+1})$.

2. Calculate the residual: $r = |D(\hat{f}(X_{n+1}))|$.

3. Check if $r \in [-\hat{q}^{\alpha}, \hat{q}^{\alpha}]$ for a given $\alpha$.

If the condition in step 3 is satisfied, the error bounds dictated by $[-\hat{q}^{\alpha}, \hat{q}^{\alpha}]$ is considered valid according to the CP framework, regardless of the specific input $X_{n+1}$.

## C. Algorithmic Procedure

1. **Set up the Neural PDE Solver**

   (a) Define the PDE system of interest with its governing equations in a numerical solver

   (b) Train a neural network (e.g., Fourier Neural Operator) to approximate solutions to the PDE

   (c) Ensure the model can make predictions on new initial conditions / PDE coefficients

2. **Define the Physics Residual Error (PRE)**

   (a) For a given PDE case with operator D, the physics residual is defined as $|D(\hat{f}(X))|$, where $\hat{f}$ is the neural PDE prediction

   (b) Create a differential operator using finite difference stencils implemented as convolutional kernels. This operator evaluates how well the predicted solution satisfies the underlying physics

3. **Generate Calibration Set**

   (a) Sample a set of initial conditions $X_1, X_2, ..., X_n$ from the domain of interest

   (b) Run the neural PDE solver on these initial conditions to get predictions $\hat{f}(X_1), \hat{f}(X_2), ..., \hat{f}(X_n)$

   (c) Compute the PRE nonconformity scores $|D(\hat{f}(X_1))|, |D(\hat{f}(X_2))|, ..., |D(\hat{f}(X_n))|$ for each prediction

4. **Calibration Process**

   (a) For a desired confidence level 1-$\alpha$ (e.g., 90% for $\alpha = 0.1$), compute the empirical quantile:

   (b) $\hat{q}^{\alpha} = \lceil (n + 1)(1 - \alpha) \rceil / n$ quantile of $|D(\hat{f}(X_1))|, |D(\hat{f}(X_2))|, ...., |D(\hat{f}(X_n))|$

   (c) This quantile represents the threshold for the physics residual that will provide the desired coverage

5. **Apply to New Predictions**

   (a) For a new initial condition $X_{n+1}$, obtain the neural PDE prediction $\hat{f}(X_{n+1})$

   (b) Compute the physics residual $|D(\hat{f}(X_{n+1}))|$

   (c) The prediction set is defined as $\mathbb{C}^{\alpha} = [-\hat{q}^{\alpha}, \hat{q}^{\alpha}]$

   (d) If $|D(\hat{f}(X_{n+1}))|$ falls within $\mathbb{C}^{\alpha}$, the prediction is considered valid at the $1 - \alpha$ confidence level

6. **Interpretation of Results**

   (a) For marginal conformal prediction: Apply steps 1-5 cell-wise across the spatial-temporal domain to get localized error bounds

   (b) For joint conformal prediction: Calculate the supremum of the normalized residuals across the entire domain to obtain global error bounds

   (c) The width of the error bounds indicates the model's physical consistency - tighter bounds suggest better alignment with the physics

7. **Decision Framework** *(Optional)*

   (a) Accept predictions where the physics residual falls within the calibrated bounds

   (b) Reject and flag predictions where the residual exceeds the bounds, potentially routing these cases to traditional numerical solvers

   (c) This approach guarantees that, with probability at least $1 - \alpha$, the physics residual of new predictions will fall within the computed bounds, ensuring that the model maintains physical consistency while providing faster solutions than traditional numerical methods.

# D. ConvOperator: Convolutional Kernels for Gradient Estimation

Within the code base for this paper, we release a utility function that constructs convolutional layers for gradient estimation based on your choice of order of differentiation and Taylor approximation. This allows for the PRE score function to be easily expressed in a single line of code [2]

This section provides an overview of the code implementation and algorithm for estimating the PRE using Convolution operations. We'll use an arbitrary PDE example with a temporal gradient $\frac{\partial u}{\partial t}$ and a Laplacian $\left( \frac{\partial^2}{\partial x^2} + \frac{\partial^2}{\partial y^2} \right)$ to illustrate the process.

$$\frac{\partial u}{\partial t} - \alpha \left( \frac{\partial^2 u}{\partial x^2} + \frac{\partial^2 u}{\partial y^2} \right) + \beta u = 0, \qquad (25)$$

where $u$ is the field variable, $t$ is time, $x$ and $y$ are spatial coordinates, and $\alpha$ and $\beta$ are constants. To estimate the PDE residual given by Equation (25), we need to estimate the associated spatio-temporal gradients.

First, we use the `ConvOperator` class from `Utils/ConvOps_2d.py` to set up the convolutional layer with kernels [3] taken from the appropriate finite difference stencils:

```
from ConvOps_2d import ConvOperator

# Define each operator within the PDE
D_t = ConvOperator(domain='t', order=1)
 #time-derivative
D_xx_yy = ConvOperator(domain=('x','y')
, order=2) #Laplacian
D_identity = ConvOperator() #Identity
Operator
```

The `ConvOperator` class is used to set up a gradient operation. It takes in `variable(s) of differentiation` and `order of differentiation` as arguments to design the appropriate forward difference stencil and then sets up a convolutional layer with the stencil as the kernel. Under the hood, the class will take care of devising a 3D convolutional layer, and setup the kernel so that it acts on a spatio-temporal tensor of dimensionality: *[BS, Nt, Nx, Ny]* which expands to batch size, temporal discretisation and the spatial discretisation in $x$ and $y$.

---

[2]The code and associated utility functions can be found in: https://github.com/gitvicky/CP-PRE

[3]Convolution functions can be set as cross-correlations as it is default in the PyTorch framework, or they could be set up as convolutions by obtaining the complex conjugate in the frequency space.(Jordà et al., 2019)

```
alpha, beta = 1.0, 0.5  # Example
coefficients
D = ConvOperator() #Additive Kernels
D.kernel = D_t.kernel − alpha * D_xx_yy
.kernel − beta * D_identity.kernel
```

The convolutional kernels are additive i.e. in order to estimate the residual in one convolutional operation, they could be added together to form a composite kernel that characterises the entire PDE residual.

Once having set up the kernels, PRE estimation is as simple as passing the composite class instance $D$ the predictions from the neural PDE surroga te (ensuring that the output is in the same order as the kernel outlined above).

```
y_pred = model(X)
PRE = D(y_pred)
```

Only operating on the outputs, this method of PRE estimation is memory efficient, computationally cheap and with the `ConvOperator` evaluating the PDE residual can be done in a single line of code.

## D.1. Impact of Discretisation

As demonstrated in (Bartolucci et al., 2023), the discretisation of the inputs and hence model outputs plays an important role in the accuracy of the neural-PDE solvers. Though the neural operators are constructed for discretisation-invariant behaviour due to the band-limited nature of the functions, they often exhibit discretisation-convergent behaviour as opposed to discretisation-invariance. This is of particular importance in the temporal dimensions as these neural-PDE models utilise a discrete, autoregressive based time-stepping, baked into the model within its training regime (McCabe et al., 2023). The lack of control in teh temporal discretisation ($dt$), leads to higher numerical errors within the the PRE estimates. In fig. 7, we visualise the evaluation of finite difference in 2D+time as a 3D convolution. The finite difference stencil i.e. the convolutional kernel has a unit discretisation of $dx$, $dy$ and $dt$ associated with the problem and is applied over the signal i.e. the output from the neural-PDE $u$ spanning the domain $x, y, t$, where $x \in [0, X]$, $y \in [0, Y]$, $t \in [0, T]$. Though the discretisation deployed within CP-PRE stems from the neural PDE solver, it consistently delivers guaranteed coverage regardless of resolution. Even with coarser discretisation, CP-PRE remains valuable as it allows for: statistical identification of poorer fit regions (marginal formulation), highlights physically inconsistent predictions (joint formulation) and enables relative assessment of physical inconsistencies across predictions. While residuals may be inflated with coarser discretisation, the corresponding bounds reflect this inflation, preserving the relative information about physical inconsistency across a series of predictions.

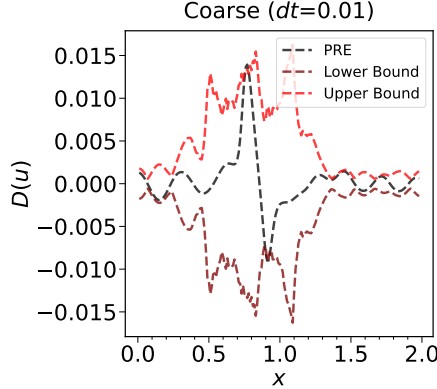

(a) CP-PRE over coarser discretisation

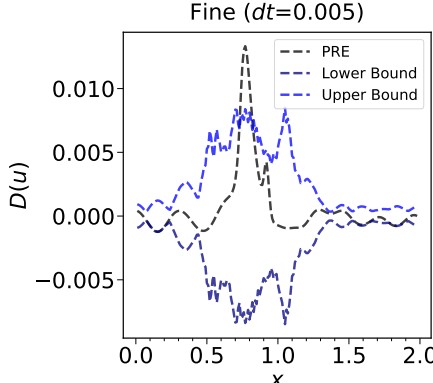

(b) CP-PRE over finer discretisation

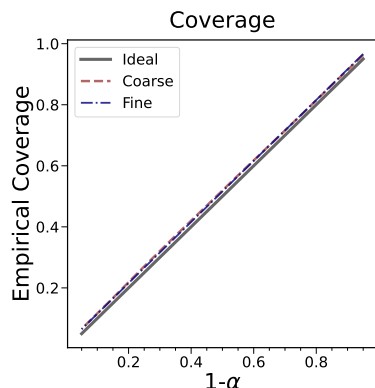

(c) Guaranteed coverage irrespective of discretisation error.

*Figure 8.* CP-PRE provides guaranteed coverage irrespective of the discretisation associated with the model outputs., however, the width of the obtained coverage bounds indicates the discretisation error associated with the gradient estimation. Coverage taken for $\alpha = 0.1 \sim 90\%$ coverage.

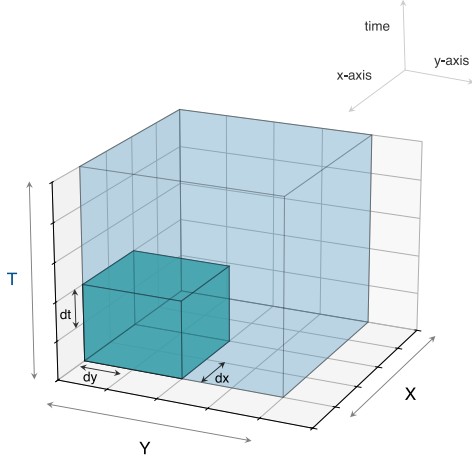

*Figure 7.* PRE estimation using the 3D convolutions with finite difference stencils as convolutional kernels being applied over the neural-PDE predictions as the signals.

## E. Initial and Boundary Conditions

As mentioned in Section 4.3, the focus of our experiments has been in quantifying the misalignment of the model with the PDE in the domain of the problem. A well-defined PDE is characterised by the PDE on the domain, the initial condition across the domain at $t = 0$ and the boundary conditions, reflecting the physics at the boundary. Within a neural-PDE setting, the initial condition does not need to be enforced or measured for as the neural-PDE is set up as an initial-value problem, taking in the initial state to autoregressively evolve the later timesteps and hence does not come under the purview of the neural-PDE's outputs. The boundary conditions, whether Dirichlet, Neumann or periodic, follows a residual structure as outlined in Equation (2), allowing us to use it as a PRE-like nonconformity score for performing conformal prediction. In all the problems we have under consideration, the PDEs are modelled under periodic boundary conditions:

$$\frac{\partial u}{\partial X} = 0; \ X \in \partial\Omega \tag{26}$$

By deploying the eqn 26 as the PRE across the boundary, we can obtain error bars over the boundary conditions as well. Within fig. 9, we demonstrate the error bars obtained by using the boundary conditions as the PRE nonconformity scores for the Navier-Stokes equations.

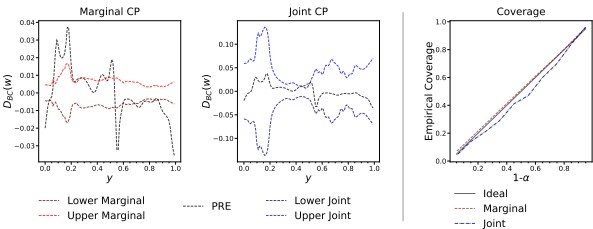

*Figure 9.* Error bars obtained over the boundary conditions over the right wall of domain of the Navier-Stokes Equation using Marginal and Joint CP. The empirical coverage obtained using the boundary condition as the PRE nonconformity score is also given.

# F. Toy Problems: 1D cases

## F.1. Advection Equation

Consider the one-dimensional advection equation

$$\frac{\partial u}{\partial t} + v\frac{\partial u}{\partial x} = 0. \tag{27}$$

The state variable of interest $u$ is bounded within the domain $x \in [0,2]$, $t \in [0,0.5]$ and moves within the domain at a constant velocity $v$. Each trajectory is generated by solving Equation (27) using a Crank-Nicolson method (Crank & Nicolson, 1947). Data is sampled using a parameterised initial condition that characterises the amplitude and position of the Gaussian field. Generated data is used to train a 1D FNO that takes in the initial condition and autoregressively with a step size of 1, learns to map the next 10 time frames as outlined in Equation (28). A reproducible script is attached to the supplementary material.

$$u^{n+1} = f(u^n), \tag{28}$$

$f$ represents the FNO and $u^n$ and $u^{n+1}$ represents the current (input) and the predicted future state of the system (output).

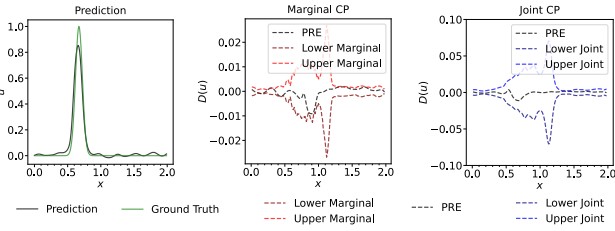

Figure 10. **Advection Equation:** (Left) Comparing the neural PDE (FNO) performance with that of the physics-based numerical solver at the last time instance. (Middle) Upper and lower bounds for 90% coverage obtained by performing marginal-CP. (Right) Upper and lower bounds for 90% coverage obtained by performing joint-CP. Marginal-CP provides tighter bounds for a prediction as opposed to joint-CP, whereas joint-CP provides a method of employing a relative sense of *reliability* of a prediction within a domain.

Figure 10 demonstrates the guaranteed bounds obtained over the residual space of Equation (27) with both the marginal and joint-CP formulations. Being cell-wise, marginal-CP guarantees coverage for each discretised point within the spatio-temporal domain. This allows for tighter bounds and error quantification of interested subdomains within regions but does not provide any guarantee across the entire prediction domain. Joint-CP acts across the entire domain and provides a guarantee as to whether a prediction (instead of a single cell) will fall within the domain or not. Larger bounds are observed as they extend over the multivariate nature of the output. Though this comes with bigger bounds,

it provides us with a mechanism to perform rejection criteria for predictions. Within joint-CP, bounds dictating $1 - \alpha$ coverage suggest that approximately, $\alpha \times 100\%$ predictions from the same domain will fall outside the bounds and can be rejected. Further details about the physics, parameterisation of the initial conditions, model and its training can be found in Appendix G.

## F.2. Burgers Equation

Consider the 1D Burgers' Equation

$$\frac{\partial u}{\partial t} + u\frac{\partial u}{\partial x} = \nu\frac{\partial^2 u}{\partial x^2}. \tag{29}$$

The state variable of interest $u$ is bounded within the domain $x \in [0,2]$, $t \in [0,1.25]$. The field is prescribed by a kinematic viscosity $\nu = 0.002$. Data is generated by solving Equation (29) using a spectral method (Canuto et al., 2007). Data sampled using a parameterised initial condition is used to train a 1D FNO that takes in the initial distribution of the state and learns to autoregressively predict the PDE evolution for the next 30 time frames.

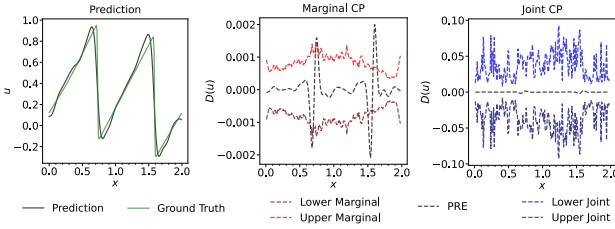

Figure 11. **Burgers' Equation:** (Left) Comparing the neural PDE (FNO) performance with that of the physics-based numerical solver at the last time instance. (Middle) Upper and lower bounds for 90% coverage obtained by performing marginal-CP. (Right) Upper and lower bounds for 90% coverage guaranteed by joint-CP over the residual space.

Figure 11 illustrates the guaranteed bounds over the residual space of Equation (29) using marginal and joint-CP formulations for 90% coverage. Marginal-CP provides cell-wise coverage, yielding tighter bounds for specific subdomains. Joint-CP provides bounds 50 times larger than that of the marginal-CP as it covers the entire prediction domain. Despite the large bounds, approximately $\alpha \times 100\%$ predictions fall outside it as given in Figure 3. For details on physics, initial condition parameterisation, model, and training, see Appendix H.

## G. 1D Advection Equation

### G.1. Physics

Consider the one-dimensional advection equation, parameterised by the initial condition:

$$\frac{\partial u}{\partial t} = vD\frac{\partial u}{\partial x}, \quad x \in [0,2], \ t \in [0,0.5],$$
$$u(x, t=0) = Ae^{(x-X)^2}. \tag{30}$$

Here $u$ defines the density of the fluid, $x$ the spatial coordinate, $t$ the temporal coordinate and $v$ the advection speed. initial condition is parameterised by $A$ and $X$, representing the amplitude and position of a Gaussian distribution. A no-flux boundary condition bounds the system.

The numerical solution for the above equation is built using a finite difference solver with a crank-nicolson method implemented in Python. We construct a dataset by performing a Latin hypercube sampling across parameters $A, X$. Each parameter is sampled from within the domain given in Table 6 to generate 100 simulation points, each with its own initial condition. Each simulation is run for 50-time iterations with a $\Delta t = 0.01$ across a spatial domain spanning [0,2], uniformly discretised into 200 spatial units in the x-axis.

*Table 6.* Domain range of initial condition parameters for the 1D advection equation.

| Parameter | Domain | Type |
|---|---|---|
| Amplitude $(A)$ | $[50, 200]$ | Continuous |
| Position $(X)$ | $[0.5, 1.0]$ | Continuous |

### G.2. Model and Training

We use a one-dimensional FNO to model the evolution of the convection-diffusion equation. The FNO learns to perform the mapping from the initial condition to the next time instance, having a step size of 1. The model autoregressively learns the evolution of the field up until the $10^{th}$ time instance. Each Fourier layer has 8 modes and a width of 16. The FNO architecture can be found in Table 7. Considering the field values governing the evolution of the advection equation are relatively small, we avoid normalisations. The model is trained for up to 100 epochs using the Adam optimiser (Kingma & Ba, 2015) with a step-decaying learning rate. The learning rate is initially set to 0.005 and scheduled to decrease by half after every 100 epochs. The model was trained using an LP-loss (Gopakumar et al., 2024b).

*Table 7.* Architecture of the 1D FNO deployed for modelling 1D Advection Equation

| Part | Layer | Output Shape |
|---|---|---|
| Input | - | (50, 1, 200, 1) |
| Lifting | Linear | (50, 1, 200, 16) |
| Fourier 1 | Fourier1d/Conv1d/Add/GELU | (50, 1, 16, 200) |
| Fourier 2 | Fourier1d/Conv1d/Add/GELU | (50, 1, 16, 200) |
| Fourier 3 | Fourier1d/Conv1d/Add/GELU | (50, 1, 16, 200) |
| Fourier 4 | Fourier1d/Conv1d/Add/GELU | (50, 1, 16, 200) |
| Fourier 5 | Fourier1d/Conv1d/Add/GELU | (50, 1, 16, 200) |
| Fourier 6 | Fourier1d/Conv1d/Add/GELU | (50, 1, 16, 200) |
| Projection 1 | Linear | (50, 1, 200, 256) |
| Projection 2 | Linear | (50, 1, 200, 1) |

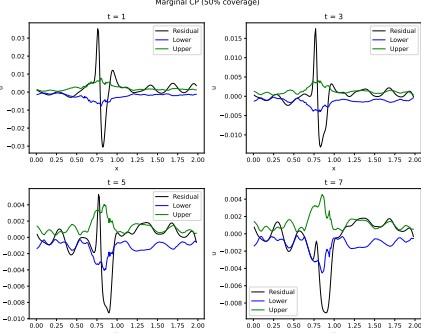

*Figure 12.* Advection Equation: Marginal-CP with $\alpha = 0.5$

### G.3. Calibration and Validation

To perform the calibration as outlined in Section 5, model predictions are obtained using initial conditions sampled from the domain given in Table 6. The same bounded domain for the initial condition parameters is used for calibration and validation. 100 initial conditions are sampled and fed to the model to obtain and prediction for both the calibration and the validation.

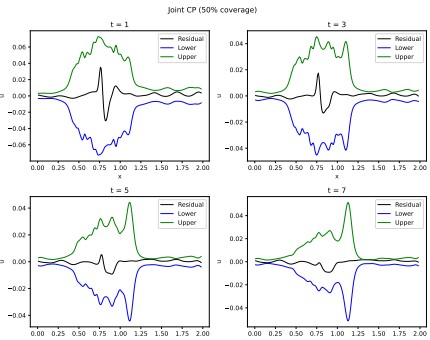

*Figure 13.* Advection Equation: joint-CP with $\alpha = 0.5$

# H. 1D Burgers Equation

## H.1. Physics

Consider the one-dimensional Burgers' equation:

$$\frac{\partial u}{\partial t} + u\frac{\partial u}{\partial x} = \nu\frac{\partial^2 u}{\partial x^2},$$

$$u(x, t = 0) = \sin(\alpha\pi x) + \cos(-\beta\pi x) + \frac{1}{\cosh(\gamma\pi x)}, \quad (31)$$

where $u$ defines the field variable, $\nu$ the kinematic viscosity, $x$ the spatial coordinate, $t$ the temporal coordinates. $\alpha, \beta$ and $\gamma$ are variables that parameterise the initial condition of the PDE setup. The system is bounded periodically within the mentioned domain.

The solution for the Burgers' equation is obtained by deploying a spectral solver (Canuto et al., 2007). The dataset is built by performing a Latin hypercube scan across the defined domain for the parameters $\alpha, \beta, \gamma$, sampled for each simulation. We generate 1000 simulation points, each one with its initial condition and use it for training.

The physics of the equation, given by the various coefficients is held constant across the dataset generation throughout as given in Equation (31). Each data point, as in each simulation is generated with a different initial condition as described above. The parameters of the initial conditions are sampled from within the domain as given in Table 8. Each simulation is run for 500-time iterations with a $\Delta t = 0.0025$ across a spatial domain spanning $[0, 2]$, uniformly discretised into 1000 spatial units in the x and y axes. The temporal domain is subsampled to factor in every $10^{th}$ time instance, while the spatial domain is downsampled to every $5^{th}$ instance.

Table 8. Domain range of initial condition parameters for the 1D Burgers' equation.

| Parameter | Domain | Type |
|---|---|---|
| $\alpha$ | $[-3, 3]$ | Continuous |
| $\beta$ | $[-3, 3]$ | Continuous |
| $\gamma$ | $[-3, 3]$ | Continuous |

## H.2. Model and Training

We train a 1D FNO to map the spatio-temporal evolution of the field variables. We deploy an auto-regressive structure that performs time rollouts allowing us to map the initial distribution recursively up until the $30^{th}$ time instance with a step size of 1. Each Fourier layer has 8 modes and a width

of 32. The FNO architecture can be found in Table 9. We employ a linear range normalisation scheme, placing the field values between -1 and 1. Each model is trained for up to 500 epochs using the Adam optimiser (Kingma & Ba, 2015) with a step-decaying learning rate. The learning rate is initially set to 0.005 and scheduled to decrease by half after every 100 epochs. The model was trained using an LP-loss (Gopakumar et al., 2024b).

Table 9. Architecture of the 1D FNO deployed for modelling 1D Burgers' equation

| Part | Layer | Output Shape |
|---|---|---|
| Input | - | (50, 1, 200, 1) |
| Lifting | Linear | (50, 1, 200, 32) |
| Fourier 1 | Fourier2d/Conv1d/Add/GELU | (50, 1, 32, 200) |
| Fourier 2 | Fourier2d/Conv1d/Add/GELU | (50, 1, 32, 200) |
| Fourier 3 | Fourier2d/Conv1d/Add/GELU | (50, 1, 32, 200) |
| Fourier 4 | Fourier2d/Conv1d/Add/GELU | (50, 1, 32, 200) |
| Fourier 5 | Fourier2d/Conv1d/Add/GELU | (50, 1, 32, 200) |
| Fourier 6 | Fourier2d/Conv1d/Add/GELU | (50, 1, 32, 200) |
| Projection 1 | Linear | (50, 1, 200, 256) |
| Projection 2 | Linear | (50, 1, 200 1) |

## H.3. Calibration and Validation

To perform the calibration as outlined in Section 5, model predictions are obtained using initial conditions sampled from the domain given in Table 8. The same bounded domain for the initial condition parameters is used for calibration and validation. 1000 initial conditions are sampled and fed to the model to perform the calibration and 100 samples are gathered for performing the validation.

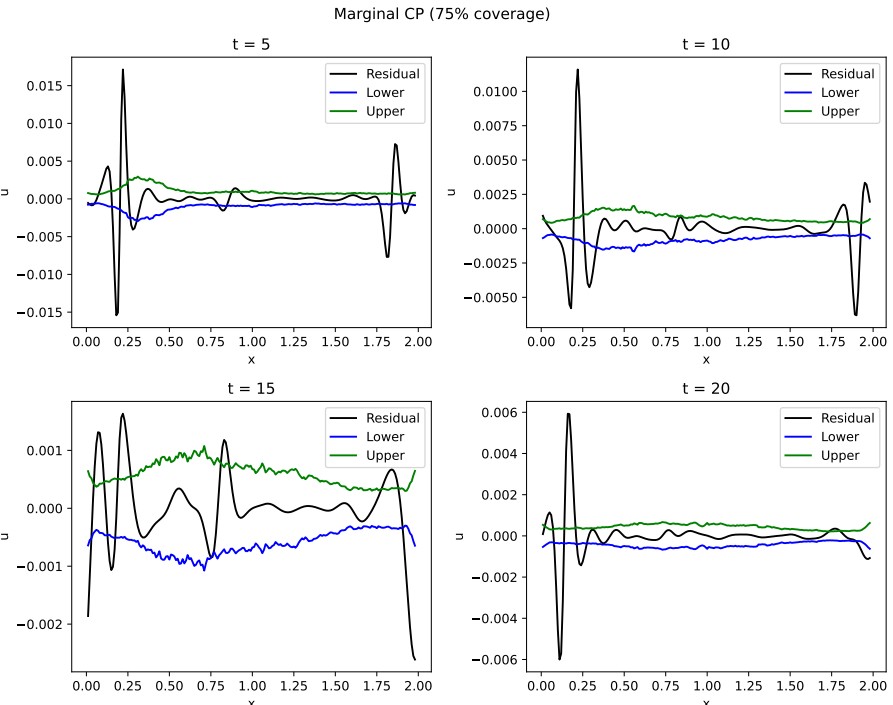

*Figure 14.* Burgers Equation: Marginal-CP with $\alpha = 0.75$

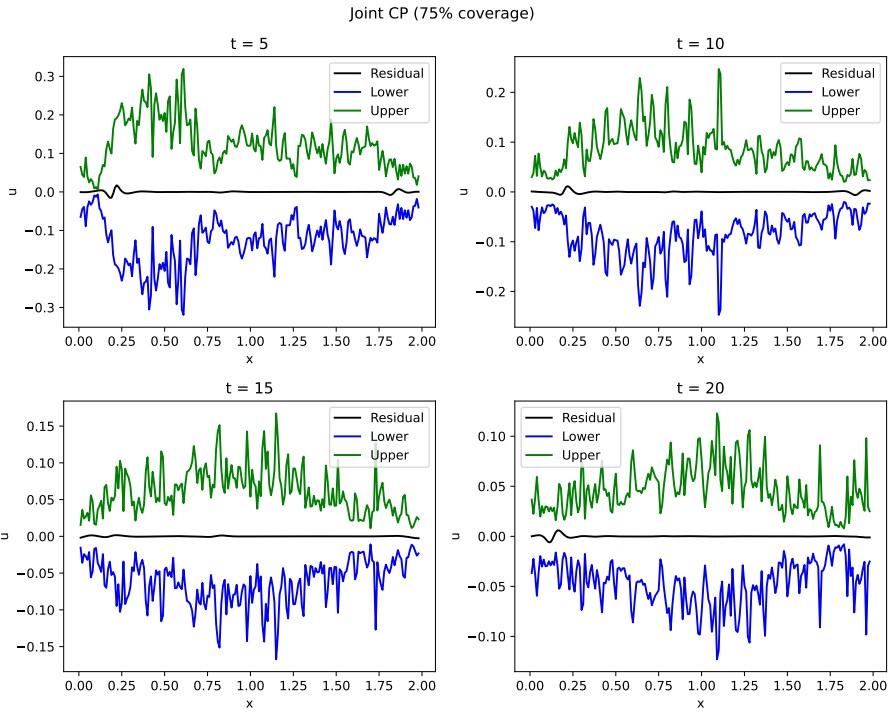

*Figure 15.* Burgers Equation: joint-CP with $\alpha = 0.75$

# I. Utilising CP-PRE as a measure of model quality

While evaluating the performance of a neural-PDE, it is important to their fit not just to the data but to the underlying physics. CP-PRE will provide guaranteed coverage irregardless of the quality of the model. It will have considerably wider error bounds when the neural-PDE (whether PINN or a Neural Operator) fails to comply with the physics. However, we believe that this is an advantage of our method. In CP-PRE formulation, the bounds are estimated across the PDE residual, where the ground truth for a well-fit solution should always be near zero. If we get wide error bars further away from the 0 for potentially high coverage estimates, it is a strong indication that statistically the model violates the physics of interest.

Consider the example with the Advection equation. We have two models, a well-fit (good model) and a poorly fit one (bad model). As shown in fig. 16, though we obtain guaranteed coverage in the case of both the bad and good models, the width of the error bars indicates the quality of the model. Taken for 90 % coverage, the width of the coverage bounds obtained over the bad model is substantially larger than that obtained by the good model.

There still could be a concern as to what width can be considered to be within a good range within the residual space. This could be estimated by running the PRE convolution operator(s) across a single numerical simulation of the interested physics, thereby estimating the impact of the operator in estimating the residual. The PRE over the simulation data will allow us to judge what ranges for the coverage width differentiate between a "bad" and a "good" model.

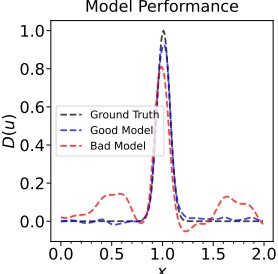

(a) Marginal advection performance

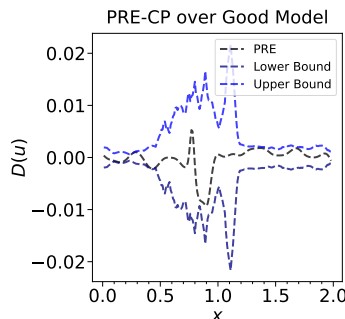

(b) Good marginal advection

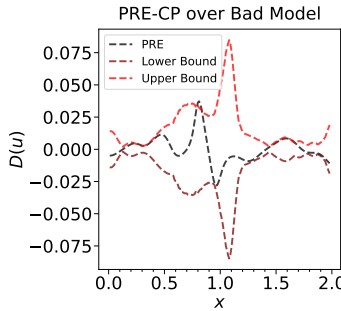

(c) Bad marginal advection

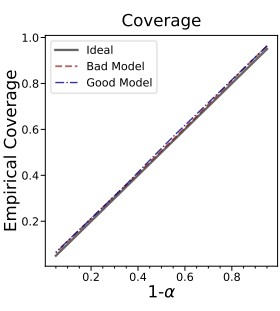

(d) Coverage analysis

*Figure 16.* CP-PRE provides guaranteed coverage irrespective of the model performance, however, the width of the obtained coverage bounds indicates the accuracy of the model in obeying the underlying physics. Coverage taken for $\alpha = 0.1 \sim 90\%$ coverage.

# J. 2D Wave Equation

## J.1. Physics

Consider the two-dimensional wave equation:

$$\frac{\partial^2 u}{\partial t^2} = c^2 \left( \frac{\partial^2 u}{\partial x^2} + \frac{\partial^2 u}{\partial y^2} \right) = 0, \quad x, y \in [-1, 1], \ t \in [0, 1],$$

$$u(x, y, t = 0) = e^{-A\left((x-X)^2 + (y-Y)^2\right)}, \tag{32}$$

$$\frac{\partial u(x, y, t = 0)}{\partial t} = 0, \qquad u(x, y, t) = 0, \quad x, y \in \partial\Omega, \ t \in [0, 1] \tag{33}$$

where $u$ defines the field variable, $c$ the wave velocity, $x$ and $y$ the spatial coordinates, $t$ the temporal coordinates. $A, X$ and $Y$ are variables that parameterise the initial condition of the PDE setup. There exists an additional constraint to the PDE setup that initialises the velocity of the wave to 0. The system is bounded periodically within the mentioned domain.

The solution for the wave equation is obtained by deploying a spectral solver that uses a leapfrog method for time discretisation and a Chebyshev spectral method on tensor product grid for spatial discretisation (Gopakumar et al., 2023a). The dataset is built by performing a Latin hypercube scan across the defined domain for the parameters $A, X, Y$, which accounts for the amplitude and the location of the 2D Gaussian. We generate 1000 simulation points, each one with its initial condition and use it for training.

The physics of the equation, given by the various coefficients, is held constant across the dataset generation throughout, as given in Equation (32). Each data point, as in each simulation is generated with a different initial condition as described above. The parameters of the initial conditions are sampled from within the domain as given in Table 10. Each simulation is run for 150-time iterations with a $\Delta t = 0.00667$ across a spatial domain spanning $[-1, 1]^2$, uniformly discretised into 64 spatial units in the x and y axes. The temporal domain is subsampled to factor in every $5^{th}$ time instance only.

*Table 10.* Domain range of initial condition parameters for the 2D wave equation.

| Parameter | Domain | Type |
|---|---|---|
| Amplitude ($A$) | $[10, 50]$ | Continuous |
| X Position ($X$) | $[0.1, 0.5]$ | Continuous |
| Y Position ($X$) | $[0.1, 0.5]$ | Continuous |

## J.2. Model and Training

We deploy an auto-regressive FNO that performs time roll-outs allowing us to map the initial distribution recursively up until the $20^{th}$ time instance with a step size of 1. Each Fourier layer has 16 modes and a width of 32. The FNO architecture can be found in Table 10. We employ a linear range normalisation scheme, placing the field values between -1 and 1. Each model is trained for up to 500 epochs using the Adam optimiser (Kingma & Ba, 2015) with a step-decaying learning rate. The learning rate is initially set to 0.005 and scheduled to decrease by half after every 100 epochs. The model was trained using an LP-loss (Gopakumar et al., 2024b). The performance of the trained model can be visualised in Figure 17.

*Table 11.* Architecture of the 2D FNO deployed for modelling the 2D wave equation

| Part | Layer | Output Shape |
|---|---|---|
| Input | - | (50, 1, 64, 64, 1) |
| Lifting | Linear | (50, 1, 64, 64, 32) |
| Fourier 1 | Fourier2d/Conv2d/Add/GELU | (50, 1, 32, 64, 64) |
| Fourier 2 | Fourier2d/Conv2d/Add/GELU | (50, 1, 32, 64, 64) |
| Fourier 3 | Fourier2d/Conv2d/Add/GELU | (50, 1, 32, 64, 64) |
| Fourier 4 | Fourier2d/Conv2d/Add/GELU | (50, 1, 32, 64, 64) |
| Fourier 5 | Fourier2d/Conv2d/Add/GELU | (50, 1, 32, 64, 64) |
| Fourier 6 | Fourier2d/Conv2d/Add/GELU | (50, 1, 32, 64, 64) |
| Projection 1 | Linear | (50, 1, 64, 64, 256) |
| Projection 2 | Linear | (50, 1, 64, 64 1) |

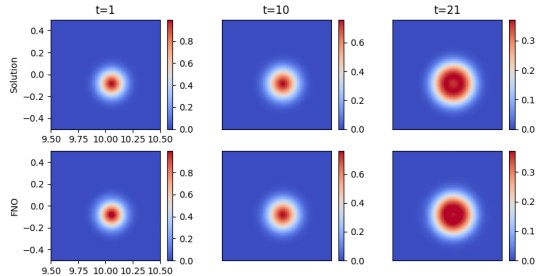

*Figure 17.* Wave Equation: Temporal evolution of field associated with the wave equation modelled using the numerical spectral solver (top of the figure) and that of the FNO (bottom of the figure). The spatial domain is given in Cartesian geometry.

## J.3. Calibration and Validation

To perform the calibration as outlined in Section 5, model predictions are obtained using initial conditions sampled from the domain given in Table 10. The same bounded domain for the initial condition parameters is used for calibration and validation. 1000 initial conditions are sampled and fed to the model to perform the calibration and 100 samples are gathered for performing the validation.

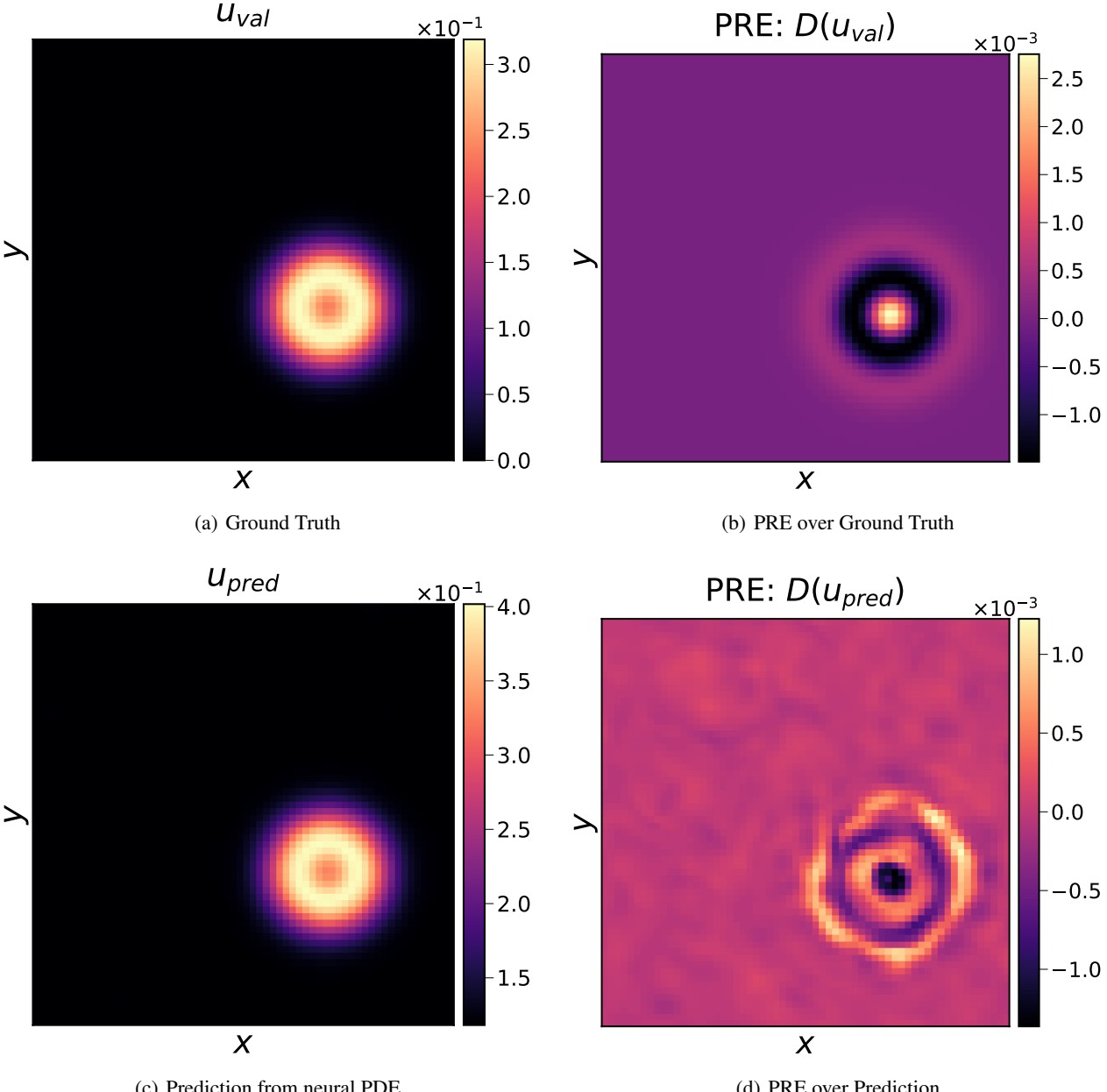

(a) Ground Truth

(b) PRE over Ground Truth

(c) Prediction from neural PDE

(d) PRE over Prediction

*Figure 18.* Analysing the PRE over the ground truth and the prediction. Though the neural PDE solver is capable of learning seemingly indistinguishable emulation of the physics while exploring the PRE over each tells a different story. As opposed to the smooth Laplacian of the PRE over the ground truth, PRE over the prediction indicates a noisy solution, potentially arising due to the stochasticity of the optimisation process.

## K. 2D Navier-Stokes Equations

### K.1. Physics

Consider the two-dimensional Navier-Stokes equations:

$$\nabla \cdot \vec{\mathbf{v}} = 0,$$

$$\frac{\partial \vec{\mathbf{v}}}{\partial t} + (\vec{\mathbf{v}} \cdot \nabla)\vec{\mathbf{v}} = \nu \nabla^2 \vec{\mathbf{v}} - \nabla P,$$

with initial conditions:

$$u(x, y, t = 0) = -\sin(2\pi\alpha y) \quad y \in [-1, 1], \tag{34}$$

$$v(x, y, t = 0) = -\sin(4\pi\beta x) \quad x \in [-1, 1], \tag{35}$$

where $u$ defines the x-component of velocity, $v$ defines the y-component of velocity. The Navier-stokes equations solve the flow of an incompressible fluid with a kinematic viscosity $\nu$. The system is bounded with periodic boundary conditions within the domain. The dataset is built by performing a Latin hypercube scan across the defined domain for the parameters $\alpha, \beta$, which parameterises the initial velocity fields for each simulation. We generate 500 simulation points, each one with its initial condition and use it for training. The solver is built using a spectral method outlined in Philip Mocz's code.

Each data point, as in each simulation is generated with a different initial condition as described above. The parameters of the initial conditions are sampled from within the domain as given in Table 12. Each simulation is run up until wallclock time reaches 0.5 $\Delta t = 0.001$. The spatial domain is uniformly discretised into 400 spatial units in the x and y axes. The temporal domain is subsampled to factor in every $10^{th}$ time instance, and the spatial domain is downsampled to factor every $4^{th}$ time instance leading to a $100 \times 100$ grid for the neural PDE.

*Table 12.* Domain range of initial condition parameters for the 2D Navier-Stokes equations

| Parameter | Domain | Type |
|---|---|---|
| Velocity x-axis ($u_0$) | $[0.5, 1.0]$ | Continuous |
| Velocity y-axis ($v_0$) | $[0.5, 1.0]$ | Continuous |

### K.2. Model and Training

We train a 2D multivariable FNO to map the spatio-temporal evolution of the field variables (Gopakumar et al., 2024b). We deploy an auto-regressive structure that performs time rollouts allowing us to map the initial distribution recursively up until the $20^{th}$ time instance with a step size of 1. Each Fourier layer has 8 modes and a width of 16. The FNO architecture can be found in Table 13. We employ a linear range normalisation scheme, placing the field values between -1 and 1. Each model is trained for up to 500 epochs using the Adam optimiser (Kingma & Ba, 2015) with a step-decaying learning rate. The learning rate is initially set to 0.005 and scheduled to decrease by half after every 100 epochs. The model was trained using an LP-loss (Gopakumar et al., 2024b). The performance of the trained model can be visualised in Figure 19.

*Table 13.* Architecture of the 2D FNO deployed for modelling 2D Navier-Stokes equations

| Part | Layer | Output Shape |
|---|---|---|
| Input | - | (50, 1, 100, 100, 1) |
| Lifting | Linear | (50, 1, 100, 100 16) |
| Fourier 1 | Fourier2d/Conv2d/Add/GELU | (50, 1, 16, 100, 100) |
| Fourier 2 | Fourier2d/Conv2d/Add/GELU | (50, 1, 16, 100, 100) |
| Fourier 3 | Fourier2d/Conv2d/Add/GELU | (50, 1, 16, 100, 100) |
| Fourier 4 | Fourier2d/Conv2d/Add/GELU | (50, 1, 16, 100, 100) |
| Fourier 5 | Fourier2d/Conv2d/Add/GELU | (50, 1, 16, 100, 100) |
| Fourier 6 | Fourier2d/Conv2d/Add/GELU | (50, 1, 16, 100, 100) |
| Projection 1 | Linear | (50, 1, 100, 100, 256) |
| Projection 2 | Linear | (50, 1, 100, 100 1) |

### K.3. Calibration and Validation

To perform the calibration as outlined in Section 5, model predictions are obtained using initial conditions sampled from the domain given in Table 12. The same bounded domain for the initial condition parameters is used for calibration and validation. 1000 initial conditions are sampled and fed to the model to perform the calibration and 100 samples are gathered for performing the validation.

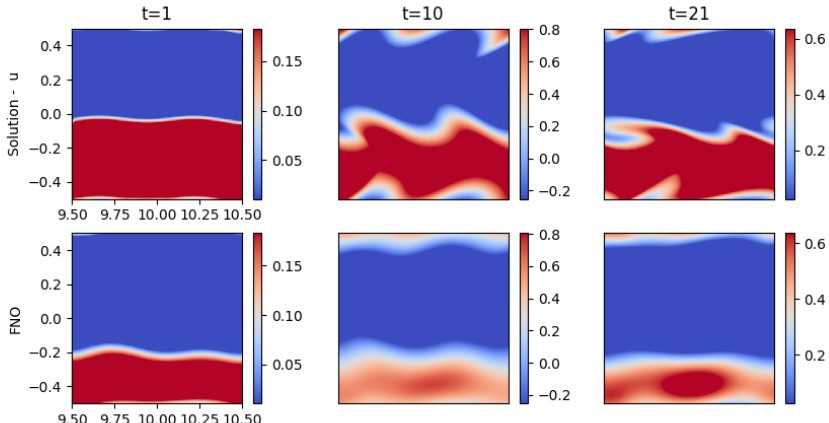

(a) Spatio-temporal evolution of the horizontal component of velocity ($u$)

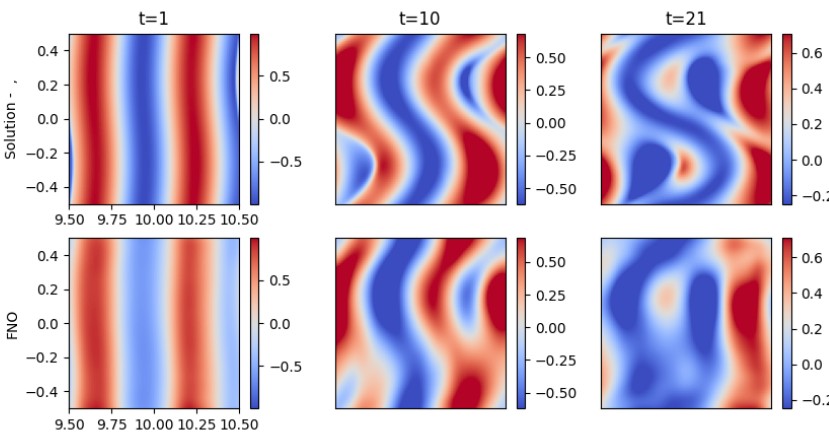

(b) Sptio-temporal evolution of the vertical component of velocity ($v$)

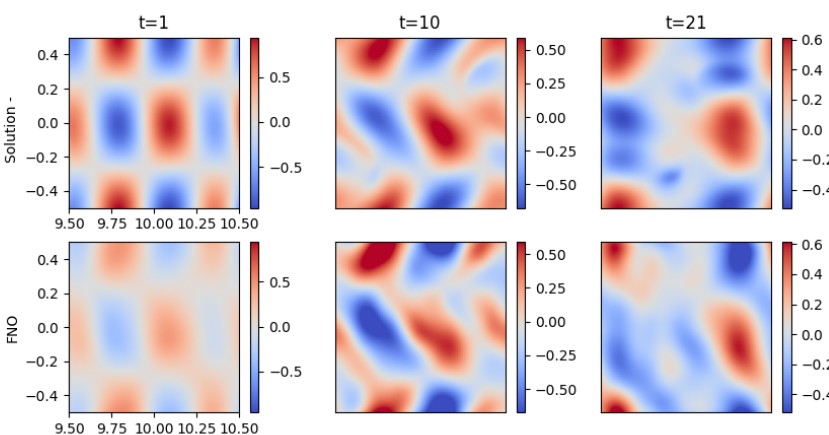

(c) Spatio-temporal evolution of the pressure field ($P$)

*Figure 19.* Navier-Stokes Equations: Temporal evolution of velocity and pressure modelled using the numerical spectral solver (top of the figure) and that of the FNO (bottom of the figure)

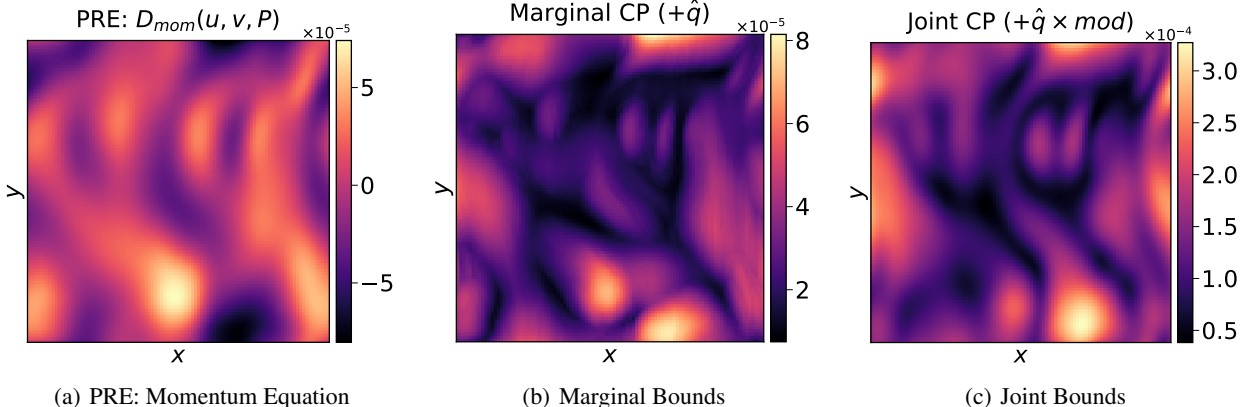

(a) PRE: Momentum Equation  (b) Marginal Bounds  (c) Joint Bounds

*Figure 20.* **Navier-Stokes:** CP using the Momentum Equation (13) as the PRE for a neural PDE surrogate model trained to model fluid dynamics. Figure 20(a) depicts the PRE, Figure 20(b) depicts the upper error bar, marginal for each cell, while Figure 20(c) indicates the upper error bar obtained across the entire prediction space. Both are estimated for 90% coverage.

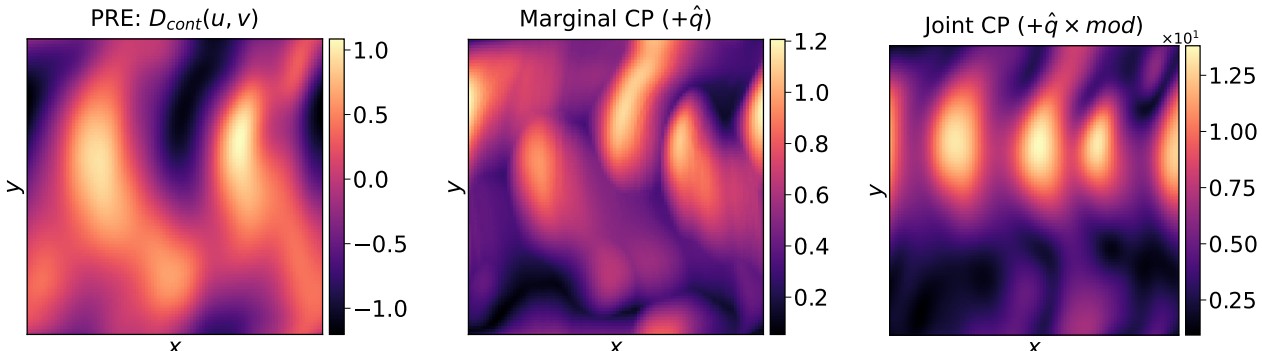

(a) PRE of the Continuity Equation over the (b) Upper error bar indicating 90% coverage (c) Upper error bar indicating 90% coverage FNO prediction with marginal-CP with joint-CP

*Figure 21.* **Navier-Stokes:** CP using the Continuity Equation (12) as the PRE for a neural PDE surrogate model trained to model fluid dynamics. Figure 20(a) depicts the PRE, Figure 20(b) depicts the upper error bar, marginal for each cell, while Figure 20(c) indicates the upper error bar obtained across the entire prediction space. Both are estimated for 90% coverage.

## L. 2D Magnetohydrodynamics

Consider the Ideal MHD equations in 2D:

$$\frac{\partial \rho}{\partial t} + \vec{\nabla} \cdot (\rho \vec{v}) = 0,$$

$$\rho \left( \frac{\partial \vec{v}}{\partial t} + \vec{v} \cdot \nabla \vec{v} \right) = \frac{1}{\mu_0} \vec{B} \times (\vec{\nabla} \times \vec{B}) - \nabla P,$$

$$\frac{d}{dt} \left( \frac{P}{\rho^\gamma} \right) = 0,$$

$$\frac{\partial \vec{B}}{\partial t} = \vec{\nabla} \times (\vec{v} \times \vec{B}),$$

$$\vec{\nabla} \cdot \vec{B} = 0,$$

with initial conditions:

$$u = -sin(2a\pi Y), \tag{36}$$

$$v = sin(2b\pi X), \tag{37}$$

$$P = \frac{\gamma}{4c\pi}, \tag{38}$$

where the density ($\rho$), velocity field ($\vec{v} = [u, v]$) and the pressure of plasma is modelled under a magnetic field ($\vec{B} = [B_x, B_y]$) across a spatio-temporal domain $x, y \in [0, 1]^2$, $t \in [0, 5]$. $\mu_0$ is taken to be the magnetic permeability of free space. The system is bounded with periodic boundary conditions within the domain. The dataset is built by performing a Latin hypercube scan across the defined domain for the parameters $a, b, c$, which parameterises the initial velocity fields for each simulation. We generate 500 simulation points, each one with its initial condition and use it for training. The solver is built using a finite volume method outlined in Philip Mocz's code.

Each data point, as in each simulation is generated with a different initial condition as described above. The parameters of the initial conditions are sampled from within the domain as given in Table 12. Each simulation is run up until wallclock time reaches 0.5 with a varying temporal discretisation. The spatial domain is uniformly discretised into 128 spatial units in the x and y axes. The temporal domain is downsampled to factor in every $25^{th}$ time instance.

*Table 14.* Domain range of initial condition parameters for the 2D MHD equations

| Parameter | Domain | Type |
|---|---|---|
| Velocity x-axis ($a$) | $[0.5, 1.0]$ | Continuous |
| Velocity y-axis ($b$) | $[0.5, 1.0]$ | Continuous |
| Pressure ($c$) | $[0.5, 1.0]$ | Continuous |

### L.1. Model and Training

We train a 2D multi-variable FNO to map the spatio-temporal evolution of the 6 field variables collectively. We deploy an auto-regressive structure that performs time rollouts allowing us to map the initial distribution recursively up until the $20^{th}$ time instance with a step size of 1. Each Fourier layer has 8 modes and a width of 16. The FNO architecture can be found in Table 15. We employ a linear range normalisation scheme, placing the field values between -1 and 1. Each model is trained for up to 500 epochs using the Adam optimiser (Kingma & Ba, 2015) with a step-decaying learning rate. The learning rate is initially set to 0.005 and scheduled to decrease by half after every 100 epochs. The model was trained using an LP-loss (Gopakumar et al., 2024b). The performance of the trained model

can be visualised in Figures 22 and 23.

*Table 15.* Architecture of the 2D FNO deployed for modelling 2D MHD equations

| Part | Layer | Output Shape |
|------|-------|--------------|
| Input | - | (50, 1, 128, 128, 1) |
| Lifting | Linear | (50, 1, 128, 128 16) |
| Fourier 1 | Fourier2d/Conv2d/Add/GELU | (50, 1, 16, 128, 128) |
| Fourier 2 | Fourier2d/Conv2d/Add/GELU | (50, 1, 16, 128, 128) |
| Fourier 3 | Fourier2d/Conv2d/Add/GELU | (50, 1, 16, 128, 128) |
| Fourier 4 | Fourier2d/Conv2d/Add/GELU | (50, 1, 16, 128, 128) |
| Fourier 5 | Fourier2d/Conv2d/Add/GELU | (50, 1, 16, 128, 128) |
| Fourier 6 | Fourier2d/Conv2d/Add/GELU | (50, 1, 16, 128, 128) |
| Projection 1 | Linear | (50, 1, 128, 128, 256) |
| Projection 2 | Linear | (50, 1, 128, 128 1) |

## L.2. Calibration and Validation

To perform the calibration as outlined in Section 5, model predictions are obtained using initial conditions sampled from the domain given in Table 12. The same bounded domain for the initial condition parameters is used for calibration and validation. 100 initial conditions are sampled and fed to the model to perform the calibration and 100 samples are gathered for validation.

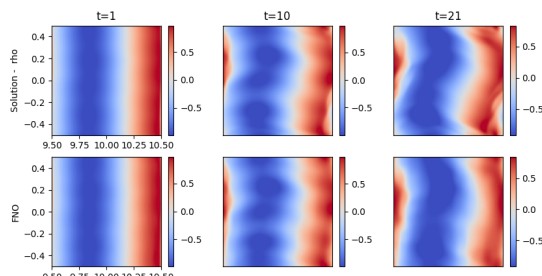

(a) Spatio-temporal evolution of density ($\rho$)

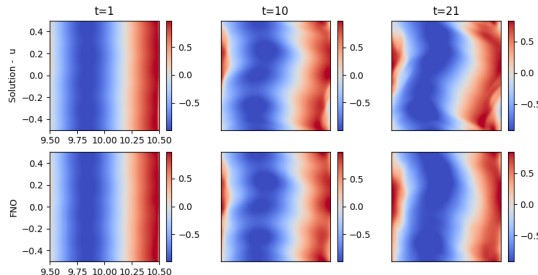

(b) Spatio-temporal evolution of the horizontal component of velocity ($u$)

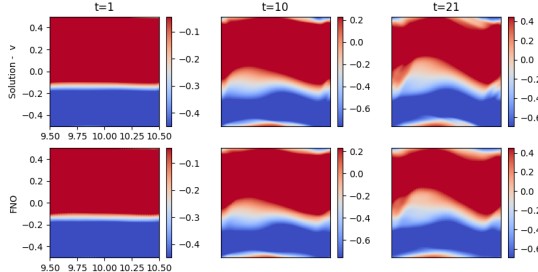

(c) Spatio-temporal evolution of the vertical component of velocity ($v$)

*Figure 22.* MHD Equations: Temporal evolution of velocity and pressure modelled using the numerical solver (top of the figure) and that of the FNO (bottom of the figure). (Continued on next page)

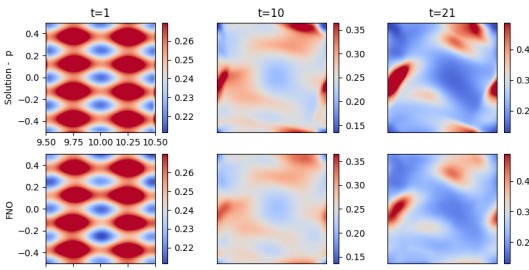

(a) Spatio-temporal evolution of the pressure field ($P$)

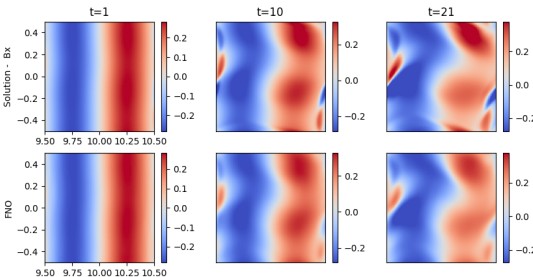

(b) Spatio-temporal evolution of the horizontal component of the magnetic field ($B_x$)

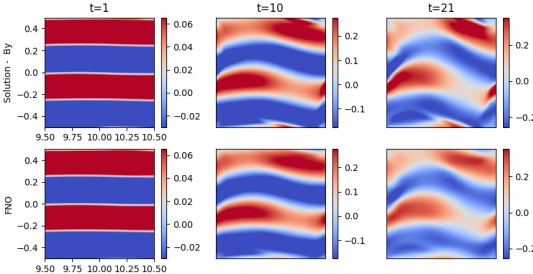

(c) Spatio-temporal evolution of the vertical component of the magnetic field ($B_y$)

*Figure 23.* MHD Equations: Temporal evolution of velocity and pressure modelled using the numerical solver (top of the figure) and that of the FNO (bottom of the figure). (Continued from previous page)

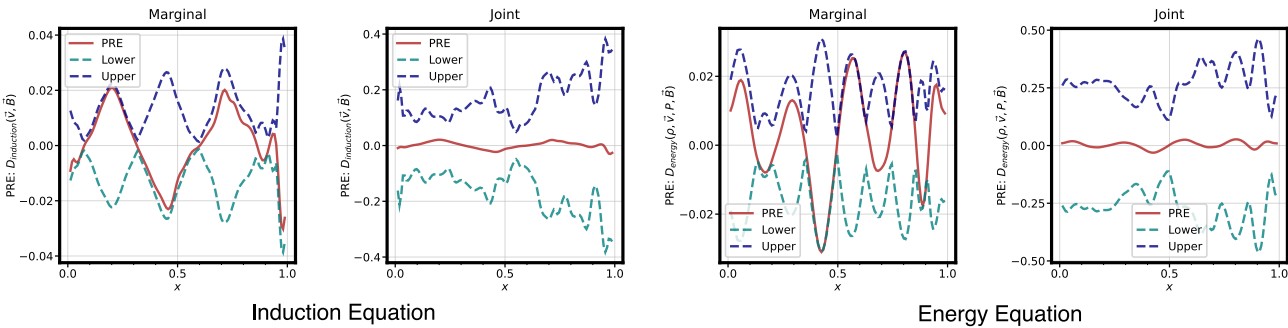

**Induction Equation**          **Energy Equation**

*Figure 24.* **MHD:** Slice plots along the x-axis (sliced at y = 0.5m) indicating the marginal and joint coverage (90%) obtained over the neural PDE modelling the MHD equations using the induction equation Equation (17) (on the left) and the energy equation Equation (16) (on the right). Marginal coverage, evaluated cell-wise, generates tight bounds to the PRE, whereas joint coverage spanning across the spatio-temporal domain introduces wider bounds.

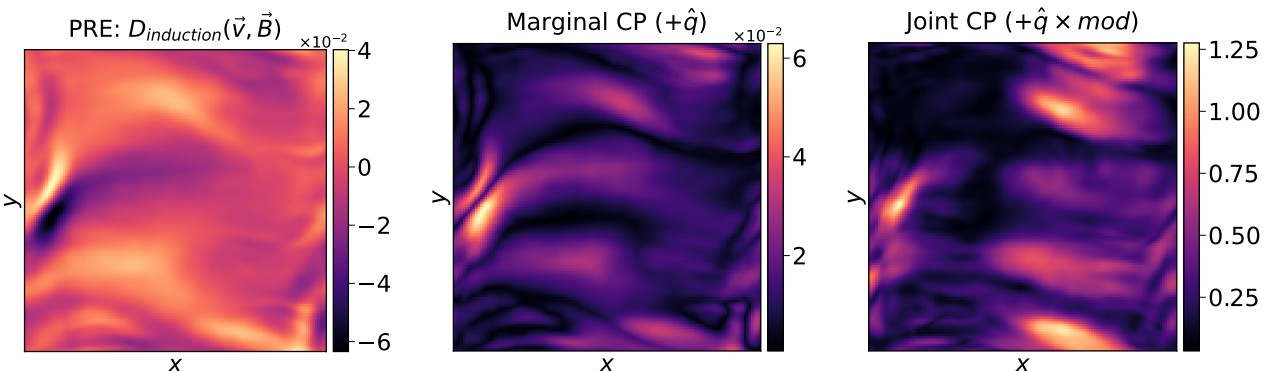

(a) PRE of the Induction Equation Equation (17) over the FNO prediction

(b) Upper error bar indicating 90% coverage with marginal-CP

(c) Upper error bar indicating 90% coverage with joint-CP

*Figure 25.* **MHD:** CP using the Induction Equation (17) as the PRE for a neural PDE surrogate model solving the Ideal MHD equations. The last time instance of the prediction is shown.

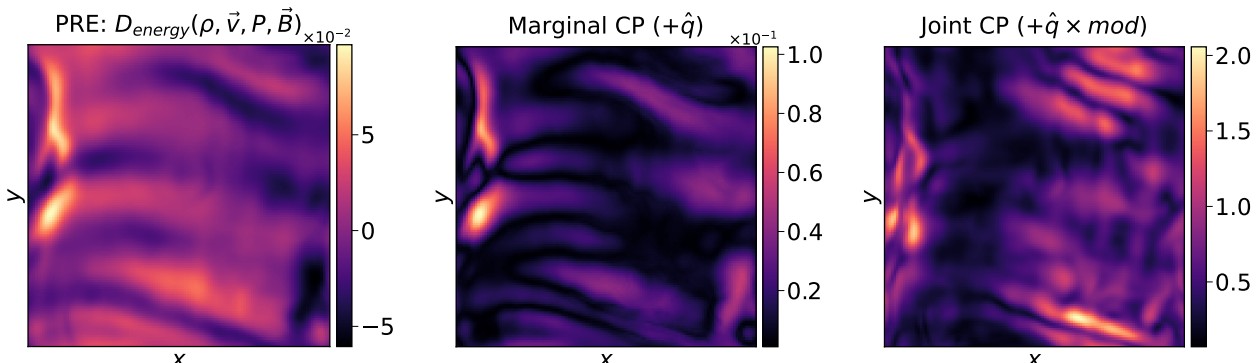

(a) PRE of the Energy Equation Equation (16) over the FNO prediction

(b) Upper error bar indicating 90% coverage with marginal-CP

(c) Upper error bar indicating 90% coverage with joint-CP

*Figure 26.* **MHD:** CP using the Energy Equation (16) as the PRE for a neural PDE surrogate model solving the Ideal MHD equations. The last time instance of the prediction is shown. Error bars obtained using joint CP are an order of magnitude higher than that obtained by marginal CP as it is measured across the entire spatio-temporal domain rather than for each cell.

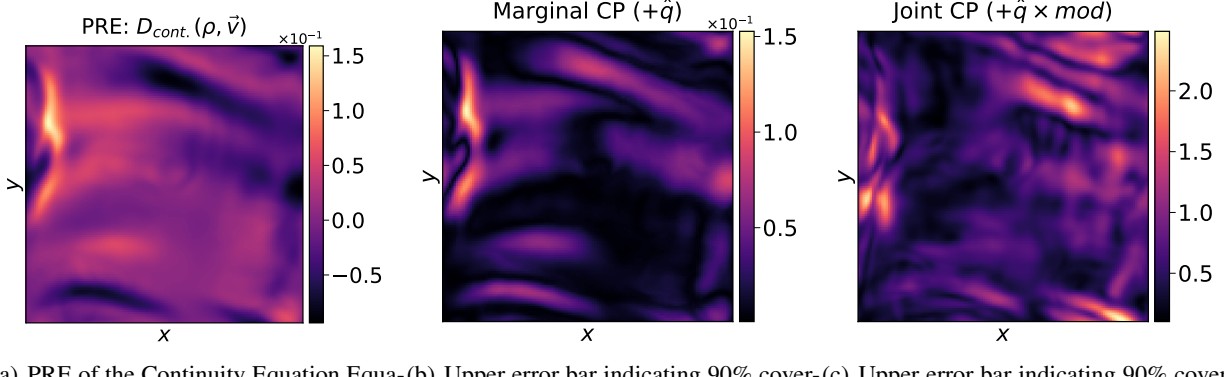

(a) PRE of the Continuity Equation Equation (14) over the FNO prediction

(b) Upper error bar indicating 90% coverage with marginal-CP

(c) Upper error bar indicating 90% coverage with joint-CP

*Figure 27.* **MHD:** CP using the Continuity Equation (14) as the PRE for a neural PDE surrogate model solving the Ideal MHD equations.

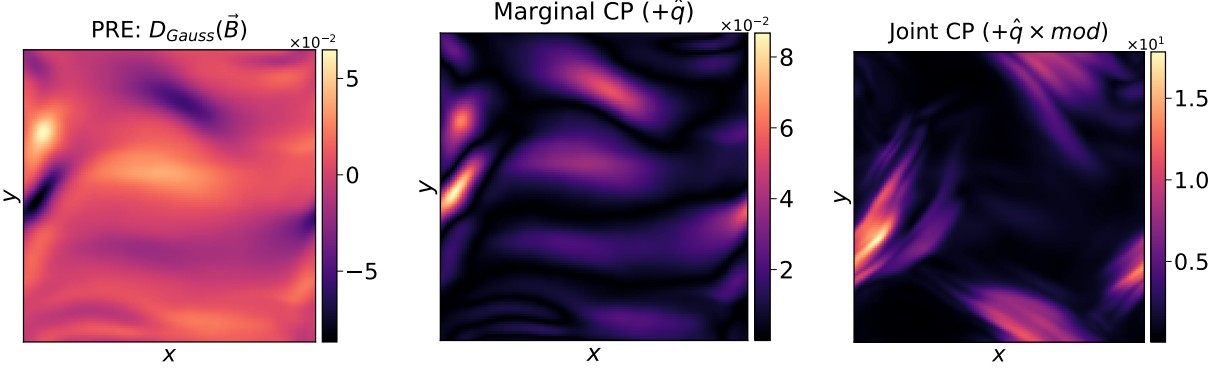

(a) PRE of the Divergence Equation Equation (18) over the FNO prediction

(b) Upper error bar indicating 90% coverage with marginal-CP

(c) Upper error bar indicating 90% coverage with joint-CP

*Figure 28.* **MHD:** CP using the Gauss's law for magnetism Equation (18) as the PRE for a neural PDE surrogate model solving the Ideal MHD equations.

# M. Plasma Modelling within a Tokamak

## M.1. Physics and Problem Setting

We evaluate our uncertainty quantification framework on a simplified Magneto-hydrodynamics (MHD) model in toroidal geometry, leveraging the dataset and neural architecture from (Gopakumar et al., 2023b). The dataset is generated using the JOREK code (Hoelzl et al., 2021) with a physics model similar to the Reduced-MHD model but with electrostatic and isothermal constraints. In this setup, only the density $\rho$, electric potential $\Phi$, and temperature $T$ fields are evolved.

The physical system models the dynamics of a toroidally axisymmetric density blob initialized on top of a low background density. Without plasma current to maintain confinement, the pressure gradient in the momentum equation acts as a buoyancy force, causing radial blob motion. This simplified scenario serves as a proxy for studying the evolution of plasma filaments and Edge-Localized Modes (ELMs) in tokamak devices.

## M.2. Dataset Generation

The dataset consists of 120 simulations (100 training, 20 testing) generated by solving the reduced MHD equations using JOREK with periodic boundary conditions. The initial conditions vary in the blob's position, width, and amplitude. Each simulation is performed within the toroidal domain with 1000 timesteps, downsampled to 100 slices and 200×200 bi-cubic finite-element spatial grid, downsampled to 100×100.

## M.3. Model Architecture and Training

We employ the Fourier Neural Operator (FNO) architecture from (Gopakumar et al., 2023b) with the following specifications:

- Input: 20 time instances of field values on 100×100 grid

- Autoregressive prediction up to 70 timesteps

- Output: Next 5 time instances

- 4 Fourier layers with width 32 and 16 modes

- Physics Normalisation followed by standard linear range normalization to [-1,1]

The training was conducted on a single A100 GPU with the Adam optimizer having an initial learning rate of 0.001 with halving every 100 epochs for 500 epochs total using a relative LP loss function. This model achieves a normalised MSE of $\sim 4e - 5$ on each field variable while providing

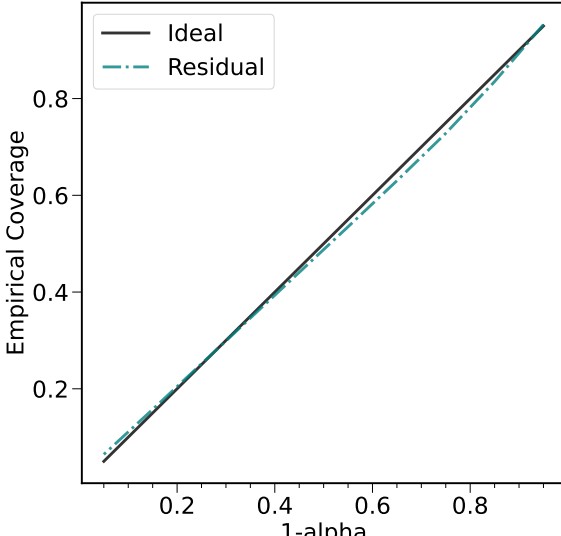

*Figure 29.* Empirical coverage obtained by performing CP-PRE over JOREK-FNO

predictions 6 orders of magnitude faster than the numerical solver. For a complete description of the MHD system and additional experimental details, we refer readers to (Gopakumar et al., 2023b).

## M.4. Calibration and Validation

For this system, we compute physics residual errors (PRE) for the temperature equation (equation 3 from (Gopakumar et al., 2024b)), as it comprises all the variables modelled by the neural PDE. Empirical coverage obtained by performing CP-PRE over the JOREK FNO is shown in Figure 29.

# N. Magnetic Equilibrium in a Tokamak

## N.1. Physics and Problem Setting

A tokamak uses magnetic fields to confine and shape a plasma for nuclear fusion. While the main toroidal field (TF) running the long way around the torus provides the primary confinement, the poloidal field (PF) coils running in loops around the cross-section are crucial for plasma stability and performance. These coils serve multiple functions: they induce the plasma current that generates an additional poloidal field component, shape the plasma cross-section, and provide vertical stability control. Without these carefully controlled poloidal fields, the plasma would be unstable and quickly lose confinement. The structure of the tokamak with emphasis on the magnetic fields and the coils are given in Figure 31.

The shape of the plasma cross-section significantly impacts performance, with modern tokamaks typically using a "D-shaped" plasma with strong elongation and triangularity. This shape, controlled actively by varying currents in different poloidal field coils, is superior to a simple circular cross-section as it allows for higher plasma current at a given magnetic field strength, improving confinement. The D-shape with triangularity also provides better stability against plasma instabilities, particularly edge localized modes, and enables better access to high-confinement operation. Maintaining this shape requires sophisticated feedback control systems since the plasma shape is naturally unstable and needs continuous adjustment, especially for maintaining the separatrix - the magnetic surface that defines the plasma boundary and creates the X-point for the divertor. A sample magnetic equilibrium across the poloidal cross-section is showcased in Figure 30, with the equilibrium shown in the contour plots and the poloidal field coils indicated with the grey block. The structure of the tokamak and the separatrix are indicated in black and red respectively.

Considering the impact of the PF coils on the equilibrium and, hence on the plasma efficiency, its placement within the structure is an open design problem for future tokamaks. Traditionally, the design space is explored by obtaining the equilibrium associated with a certain PF coil configuration, which involves solving the Grad-Shafranov equation using numerical solvers, rendering them computationally expensive. As an alternative, we construct a surrogate model to explore the design space significantly faster.

## N.2. Surrogate Model

A conditional auto-encoder, as given in Figure 32, is trained as a surrogate model that can approximate the magnetic equilibrium for a given configuration of PF coils. The auto-encoder takes in as input the spatial domain of the tokamak as the input and outputs the magnetic equilibrium

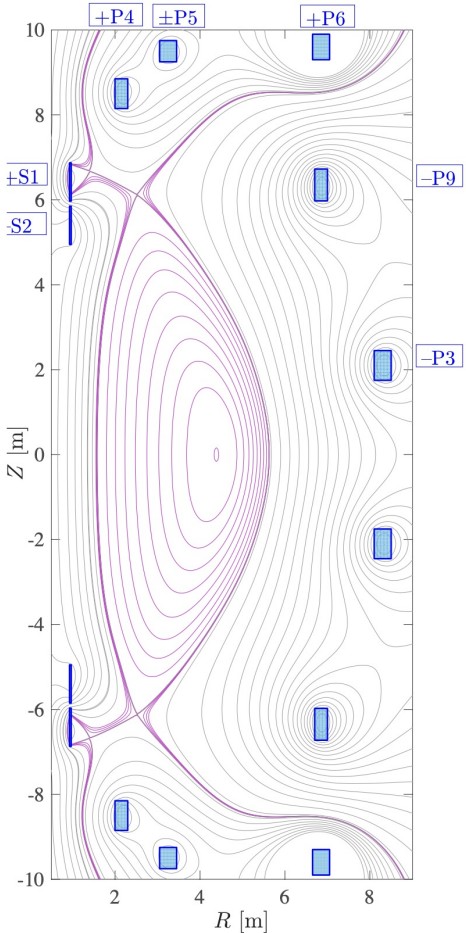

*Figure 30.* Sample Equilibrium plot showcasing the magnetic equilibrium as the contour plots observed for a given poloidal field (PF) configuration (PF coil locations are indicated in blue blocks). The poloidal cross-section of the tokamak is shown here, with the structural boundary in black and the closed flux surfaces in pink (Meyer & Team, 2024). Our problem looks at mapping the equilibrium across the spatial domain for a given PF coil location.

while being conditioned on the locations of the PF coils in the latent space. The performance of the model can be found in Figure 33. The model was trained on 512 simulations of the Grad-Shafranov simulations obtained using the FreeGSNKE code (Amorisco et al., 2024). The coil currents are kept constant as we are looking to identify the ideal coil configuration for a steady-state plasma.

## N.3. Calibration and Validation

CP-PRE is performed by evaluating the Grad-Shafranov equation over the solution space of the surrogate model, with inputs generated by sampling within the bounds of the PF coil locations. By using the GS equation, we can identify the predictions from the surrogate model that generate untenable

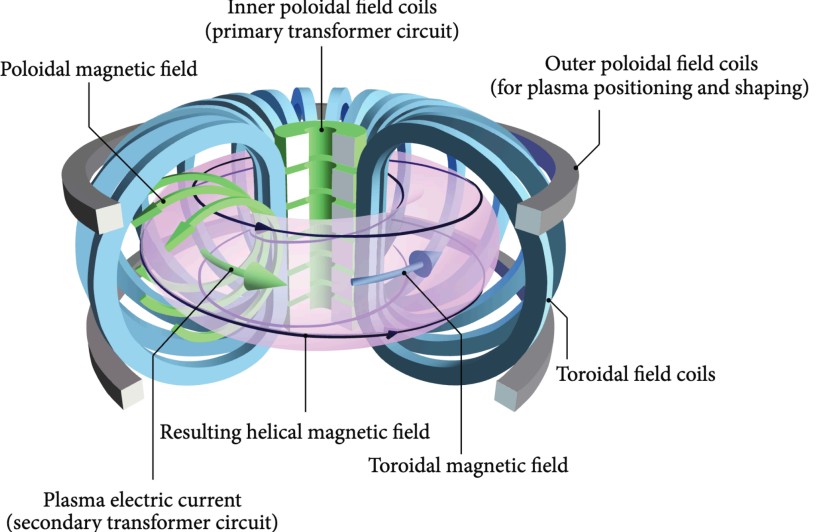

*Figure 31.* A simple schematic diagram of a generic tokamak with all of the main magnetic components and fields shown (Li et al., 2014). The poloidal field coil magnets (grey) are that which this work aims to optimise.

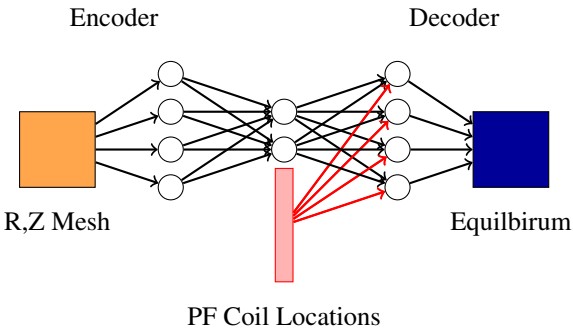

*Figure 32.* Conditional auto-encoder developed as a surrogate model for mapping the poloidal field coil locations to the corresponding magnetic equilibrium under constant coil currents.

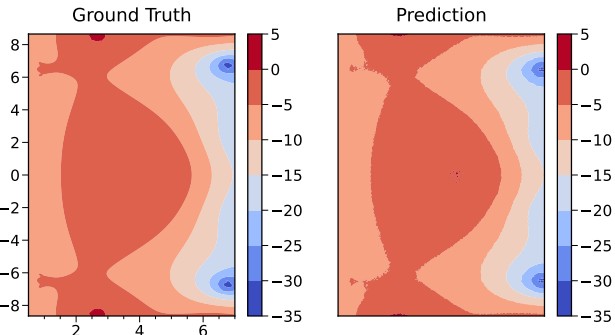

*Figure 33.* Comparing the ground truth with the prediction from the surrogate model.

equilibria. This allows us to explore the design space quickly while adding trustworthiness to your surrogate model. Both marginal and coverage obtained using CP-PRE with GS equations over the surrogate model is indicated in Figure 34. Marginal-CP shows smooth coverage as it represents the coverage averaged across each cell.

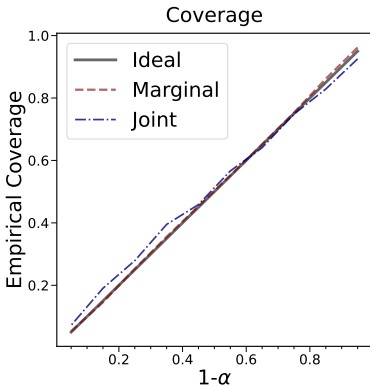

*Figure 34.* Marginal and joint empirical coverage obtained by performing CP-PRE over the Grad-Shafranov equation

