# OpenReview forum: "Calibrated Physics-Informed Uncertainty Quantification"
_ICML.cc/2025/Conference — ICML 2025 poster_

### Official Review · Reviewer_X8Gm · 2025-03-12

**Overall Recommendation:** 2

**Summary:**

The paper focuses on uncertainty quantification in physics-informed models via conformal prediction. It uses a neural network based surrogate (specifically FNO) as the base model and provides uncertainty via marginal and joint CP. The main innovation when compared to previous methods (Gopakumar et al 2024a) is the new score function that does not require simulator data ($Y_i$) and is instead using the surrogate itself to determine the error in the PDE. The method is tested on standard problems as well as a more advanced plasma modelling example following (Gopakumar et al 2024a).

**Claims And Evidence:**

The claims (as listed in lines 27-29, column 2) are supported.

**Essential References Not Discussed:**

Not that I can tell.

**Experimental Designs Or Analyses:**

The design of experiments closely follows previous studies, e.g. (Gopakumar et al 2024a) and similar literature.

**Methods And Evaluation Criteria:**

The proposed method and evaluation criteria make sense for the given problem though I have some concerns over the methodology itself (see questions below). Some benchmark examples and comparisons are provided in the supplement.

**Other Comments Or Suggestions:**

The paper is overall well-written.

Line 368: Feel - fail

**Other Strengths And Weaknesses:**

Could you discuss the results in Tables 3-5 in more detail? What is the Evaluation time, particularly in the context of the two CP based methods. As far as I understand, the CP-AER method will need to evaluation the FNO and run the simulator once for each CP sample while CP-PRE does not require the simulator but requires the various differential operators to be applied to FNO. I assume the evaluation of FNO is very fast and I can see how for the examples in your paper the simulator evaluations may also be very fast as long as an optimised solver appropriate for the given problem is used. However, I do not have a good intuition of how expensive are the evaluations of the differential operators when applied to FNO but I would guess they are not much better than the simulator since they are performing ultimately the same calculations and the simulator may be highly optimised (e.g. FEM using high performance linear algebra libraries). Can you comment more on this and hence the resulting evaluation times? Can you then provide some guidance on when CP-AER is preferred over CP-PRE?

The results shown in the main part of the paper are hard to interpret. For example, looking at Fig. 4 or Fig. 8, it is not obvious whether the method does well or poorly, or how the CP uncertainty estimates could be interpreted. This is further obscured by the fact that the predictions of the model are not in the original parameter/observation space but it in the residual space, as also pointed out by the authors. A broader discussion of the results, their interpretation and the implications would make the Experimental section stronger. I would also suggest replacing some of the figures in the main part of the paper with the model comparison results that are currently given in Supplement C, e.g. Table 2 and Tables 3-5. I believe these offer more insight than the visual representations of the solutions.

**Questions For Authors:**

The primary area for this paper is  Applications->Chemistry, Physics, and Earth Sciences. However, the problems shown in the paper are closely aligned with examples used in existing literature and do not solve a new problem (as far as I can tell). Is there a particular reason you chose this primary area?

**Relation To Broader Scientific Literature:**

Yes, the key contributions are related to the broader literature. Table 2 in Supplement C provides a high level overview of how this paper fits within the broader literature and some discussion on relevant literature in both Physics-informed ML and CP is given throughout the paper.

**Theoretical Claims:**

I did not check every step in the theoretical results (Supplement A) but I believe the results follows from standard CP results as long as the residuals are exchangeable.

---

> ### Author Rebuttal · Authors · 2025-03-31
>
> # Evaluation times for CP-AER and CP-PRE:
> Thank you for highlighting the need for clarity regarding computational costs. We've updated the table to separately report calibration times for both methods:
>
> | PDE  | UQ                           | L2 \(ID\)                    | Coverage \(ID\)     | L2 \(OOD\)                   | Coverage\(OOD\)     | Train time \(hr\) | Eval\. time \(s\) | Cal\. time \(s\) |
> |:----:|:----------------------------:|:----------------------------:|:-------------------:|:----------------------------:|:-------------------:|:-----------------:|:-----------------:|:----------------:|
> | Wave | CP\-AER                      | 1\.76e\-05 $\\pm$ 4\.40e\-07 | 95\.70 $\\pm$ 0\.21 | 2\.46e\-03 $\\pm$ 1\.41e\-05 | 95\.59 $\\pm$ 0\.14 | 0:38              | 22                | 2000             |
> |      | **CP\-PRE \(Ours\)** | 1\.78e\-05 $\\pm$ 4\.61e\-07 | 95\.52 $\\pm$ 0\.21 | 2\.46e\-03 $\\pm$ 1\.25e\-05 | 95\.39 $\\pm$ 0\.12 | 0:38              | 22                | **10**   |
> | NS   | CP\-AER                      | 1\.05e\-04 $\\pm$ 6\.58e\-06 | 95\.56 $\\pm$ 0\.40 | 3\.66e\-03 $\\pm$ 2\.81e\-05 | 95\.54 $\\pm$ 0\.15 | 3:22              | 25                | 20000            |
> |      | **CP\-PRE \(Ours\)** | 1\.07e\-04 $\\pm$ 5\.18e\-06 | 95\.44 $\\pm$ 0\.22 | 3\.70e\-03 $\\pm$ 4\.23e\-05 | 95\.57 $\\pm$ 0\.14 | 3:22              | 25                | **100**  |
> | MHD  | CP\-AER                      | 2\.20e\-03 $\\pm$ 4\.38e\-05 | 95\.61 $\\pm$ 0\.26 | 4\.69e\-02 $\\pm$ 8\.18e\-04 | 95\.60 $\\pm$ 0\.27 | 5:00              | 40                | 30000            |
> |      | **CP\-PRE \(Ours\)** | 2\.20e\-03 $\\pm$ 4\.96e\-03 | 95\.54 $\\pm$ 0\.18 | 4\.71e\-02 $\\pm$ 1\.06e\-03 | 95\.67 $\\pm$ 0\.22 | 5:00              | 40                | **400**  |
>
>
> Evaluation (Eval.) time refers to the time taken to evaluate the FNO. Calibration (Cal.) times refer to the time for data generation (CP-AER) of 1000 simulations or residual estimation (CP-PRE) over 1000 predictions. All evaluation and calibration times are done on a standard laptop, while the training is done a single A100 GPU.
>
> CP-AER requires substantial computational resources for calibration data generation, often through complex finite volume/element simulations that demand domain-specific expertise. CP-PRE leverages finite difference stencils as convolutional kernels, enabling GPU-parallelized computation through ML libraries with simultaneous space-time evaluation and cross-domain transferability. Our additive operator structure balances computational efficiency with statistical sufficiency (Appendix D).
>
> ***
> ***
> # Guidance on when CP-AER is preferred over CP-PRE:
> Thank you for this important question. Each method offers distinct advantages depending on the application context:
>
> CP-AER is preferred when:
>
> •	Physics knowledge is incomplete or uncertain
>
> •	Bounds on physical vector fields are specifically needed
>
> •	Sufficient computational budget exists for calibration data
>
> •	The system has unknown or complex physics, unable to be expressed in a residual form
>
> CP-PRE is advantageous when:
>
> •	Calibration data is prohibitively expensive
>
> •	Complete physics knowledge is available to get bounds on conservative variables
>
> •	Computational efficiency is critical
>
> The primary limitation of CP-AER is its dependence on calibration data, which becomes expensive as simulation complexity grows. Meanwhile, CP-PRE only requires sufficient physics knowledge to formulate the equality constraint needed for the residual calculation.
>
>
> ***
> ***
> # Broader discussion of the results and their implications:
> We thank the reviewer for this suggestion and upon much deliberation we agree that it’s more informative to provide the tables in demonstrating coverage as opposed to the figures of the residuals and the bounds. The tables (2-5) help illustrate our idea both qualitatively and quantitatively without needing any domain knowledge. The figures showing the bounds over the physical fields are indicative of the conservation laws associated with the problem and might not provide clarity to the uninitiated reader. We have the meaning of the figures in detail for the same query raised by reviewer MQ8g (under utility of method). We have restructured the paper to abide by your suggestion. Thank you for your valuable advice.
>
> ***
> ***
> # Primary area as Applications->Chemistry, Physics, and Earth Sciences:
> Our primary area selection reflects the paper's core contribution to physics and computational science applications. While using established benchmarks, our framework addresses a significant gap in scientific computing by providing statistically guaranteed, physics-informed uncertainty quantification for neural PDE surrogates.
> Sections 5.4-5.5 demonstrate applications to fusion plasma modelling and tokamak magnetic equilibrium, showing how our method enhances reliability in high-consequence scientific domains with rigorous uncertainty bounds.

---

> > ### Comment · Reviewer_X8Gm · 2025-04-05
> >
> > Thank you for the response. While I appreciate the clarifications, the utility of the proposed methodology appears limited (requiring good knowledge of the physics of the problem and ability to implement data driven solvers while offering only minor improvements on existing methods).

---

> > > ### Author Response · Authors · 2025-04-07
> > >
> > > ## UQ for downstream deployment of Neural PDE solvers
> > >
> > > We respectfully disagree that our method has limited utility. **Knowing the underlying physics is precisely the scenario where our method provides unique value.**
> > >
> > > Neural PDE solvers (like traditional numerical solvers) are specifically designed for scenarios where the governing equations are known—whether they're physics-informed neural networks [1], neural operators handling families of solutions [2, 3], or foundation models for physics [4, 5, 6]. The critical gap in this field is not solving PDEs but quantifying the trustworthiness of these solutions with statistical guarantees (as illustrated in figure 1 and explained further in the rebuttal to reviewer MQ8g under the big-picture). Our PRE-CP framework addresses this by providing:
> > >
> > > Our PRE-CP framework provides unique value through **model-agnostic calibration** that works as a post-hoc measure on any trained neural PDE solver without architectural modifications, while delivering the first **physics-informed coverage guarantees** for neural PDEs essential for high-stakes applications; it requires **no ground truth data for calibration** (applicable when training data is unavailable/proprietary), and offers **decision-enabling information** through a joint-CP formulation that determines when predictions likely violate physical laws—creating a principled basis for choosing between neural solvers and traditional numerical methods.
> > >
> > > In domains like climate prediction, fusion plasma control, or aerospace design, neural PDE solvers could provide 1000× speedups but aren't deployed due to uncertainty concerns. Our method enables these applications with calibrated uncertainty bounds that respect physical laws.
> > >
> > > The reviewer mentions "minor improvements over existing methods," but we emphasise that no existing method provides statistically valid, physics-informed uncertainty quantification for neural PDEs without requiring data, model modifications, or sampling. **In PDE-based applications, the governing physics equations are known a priori—the exact forward problem setting where our method excels**. As neural solvers become increasingly accessible as black-box tools (such as emerging foundation models), our approach uniquely bridges this critical gap, adding solution reliability to uncalibrated neural PDEs.
> > >
> > > ***
> > > ***
> > > ## Further applications of PRE-CP
> > >
> > > Our framework applies to any prediction case where the forward problem can be formulated as a residual with an equality constraint (Appendix A, B). The framework works in any scenario where the forward model can be expressed in the standard canonical form:
> > >
> > > $$ \mathcal{D}(u) - b = 0$$
> > >
> > > Where $u$ is the model prediction, $\mathcal{D}$ is a differential or algebraic operator governing the dynamics, and $b$ is a non-homogenous term such as a function or a constant.
> > >
> > > This extends our applications beyond PDEs to ODEs and algebraic equations found in control problems [7], chemical reactions [8], biological systems [9], and financial scenarios [10].
> > >
> > > We deliberately focus on PDEs as they represent the most comprehensive and challenging case—multi-dimensional domains with complex spatio-temporal dependencies and unique computational challenges. Success with PDEs implicitly validates applicability to simpler systems.
> > >
> > > We're currently extending our PRE-CP framework as an acquisition function within an active learning pipeline, as data generation from complex simulation codes is computationally intensive. This method could provide a data-free, model-agnostic, physics-informed approach to sampling training data from expensive simulation codes [11].
> > >
> > > ---
> > >
> > > **References**
> > >
> > > [1] Raissi, M., et al. (2019). Physics-informed neural networks. Journal of Computational Physics, 378, 686-707.
> > >
> > > [2] Kovachki, N., et al. (2023). Neural Operator: Learning Maps Between Function Spaces. JMLR, 24(89), 1-97.
> > >
> > > [3] Lu, L., et al. (2021). Learning nonlinear operators via DeepONet. Nature Machine Intelligence, 3(3), 218-229.
> > >
> > > [4] McCabe, M., et al. (2024). Multiple Physics Pretraining for Physical Surrogate Models. arXiv:2310.02994.
> > >
> > > [5] Rahman, M. A., et al. (2024). Pretraining Codomain Attention Neural Operators for Multiphysics PDEs. arXiv:2403.12553.
> > >
> > > [6] Herde, M., et al. (2024). Poseidon: Efficient Foundation Models for PDEs. arXiv:2405.19101.
> > >
> > > [7] Jiang Y, Jiang ZP. (2014). Robust adaptive dynamic programming and feedback stabilization. IEEE Trans Neural Netw Learn Syst, 25(5), 882-93.
> > >
> > > [8] Thöni, A. C. M., et al. (2025). Modelling Chemical Reaction Networks using Neural ODEs. arXiv:2502.19397.
> > >
> > > [9] Wang, S., et al. (2019). Massive computational acceleration using neural networks to emulate mechanism-based biological models. Nature Communications, 10, 4354.
> > >
> > > [10] Liu, S., et al. (2019). Pricing Options and Computing Implied Volatilities using Neural Networks. Risks, 7(1), 16.
> > >
> > > [11] Musekamp, D., et al. (2025). Active Learning for Neural PDE Solvers. Proc. ICLR 2025.

---

### Official Review · Reviewer_Qd74 · 2025-03-13

**Overall Recommendation:** 3

**Summary:**

This paper presents a method for estimating uncertainties in neural PDE solvers without requiring labeled data. The authors propose PRE-CP, which combines PDE residuals and conformal prediction. By using the PDE’s own equations as the reference, the method calibrates each model’s physical errors directly. They show that PRE-CP works with standard neural PDE setups (such as wave, Navier-Stokes, and magnetohydrodynamics) and also demonstrate applications to fusion research. Their results indicate that PRE-CP can uncover locations in the model’s predictions where the solution fails to follow the underlying physics, offering a statistically valid way to determine when to trust or question the model’s output.

## update after rebuttal
Rebuttal acknowledged.

**Claims And Evidence:**

The key claim is that using a physics-informed residual as a nonconformity score leads to valid coverage guarantees. The paper supports this with theoretical arguments and empirical validation on multiple PDEs.

**Essential References Not Discussed:**

Most relevant PDE-based operator learning work is cited (Fourier Neural Operators, Physics-Informed Neural Networks), and the paper mentions conformal prediction references for spatio-temporal data.

**Experimental Designs Or Analyses:**

The experiments compare coverage results across wave, Navier-Stokes, and magnetohydrodynamics PDEs, and also show real-world scenarios. The analyses use marginal and joint coverage and show how coverage is empirically measured. The experimental protocols appear logically consistent, with error bars and coverage curves displayed clearly.

**Methods And Evaluation Criteria:**

The authors employ standard PDE examples (wave, Navier-Stokes, MHD) and then apply the approach to practical fusion applications. The authors use recognized PDE solvers for reference. They also describe how to generate PDE residuals efficiently via finite difference stencils

**Other Comments Or Suggestions:**

+ README for Code: Including a short README in the supplementary code would help others replicate the experiments.
+ Extended Discretization Analysis: More discussion about how changing resolution in space/time affects the residual estimates would be helpful.

**Other Strengths And Weaknesses:**

Strengths:
+ It provides both marginal and joint coverage formulations, giving users the option to pinpoint local errors or reject predictions at a global scale.
+ It provides coverage guarantees without labeled data.
+ The appendices are thorough, showing proofs for the theoretical aspects and comparisons across multiple UQ methods and PDE scenarios.

Weaknesses：
+ The approach requires knowing the PDE precisely, so it is not suitable if the physical model is partially unknown or has uncertain terms.
+ The residual-based method can be sensitive to discretization, especially in the temporal domain, potentially inflating error bars in coarser meshes.
+ Initial and boundary conditions, while addressed, could receive a clearer explanation in the main text for completeness.

**Questions For Authors:**

+ If certain PDE terms or coefficients are unknown, could this approach handle partially specified physics by defining an approximate residual operator?
+ How large is the overhead when evaluating PDE residuals for high-resolution spatio-temporal domains, and are there workarounds to reduce it without losing coverage guarantees?

**Relation To Broader Scientific Literature:**

This work expands conformal prediction into PDE-solving contexts by incorporating PDE residuals. It aligns with recent research on physics-informed methods for PDE-based modeling (e.g., PINNs, neural operators). Unlike many UQ approaches requiring labels or Bayesian sampling, this paper uses physics-based residuals, filling a gap in data-free calibration methods. It also relates to prior CP frameworks for high-dimensional data, extending them to spatio-temporal PDE outputs.

**Theoretical Claims:**

The proof sketches in the appendices appear coherent. A deeper check would require verifying each step of their derivation regarding exchangeability assumptions for PDE initial conditions, but no immediate flaws are evident.

---

> ### Author Rebuttal · Authors · 2025-03-31
>
> # README for Code:
>
> Thank you for pointing out this error that happened during anonymisation and giving us a chance to rectify it. For the purpose of the review, we are providing an abridged README below.
>
> ## Installation
>
> ```bash
> pip install -r requirements.txt
> ```
>
> ## Quick Start
>
> Run standalone experiments (no data or pre-trained models needed):
>
> ```bash
> # Marginal bounds for 1D advection
> python -m Marginal.Advection_Residuals_CP
>
> # Joint bounds for 1D advection
> python -m Joint.Advection_Residuals_CP
> ```
>
> ## PRE Estimation Example
>
> ```python
> from ConvOps_2d import ConvOperator
>
> # Define operators for PDE: ∂u/∂t - α(∂²u/∂x² + ∂²u/∂y²) + βu = 0
> D_t = ConvOperator(domain='t', order=1)  # time-derivative
> D_xx_yy = ConvOperator(domain=('x','y'), order=2)  # Laplacian
> D_identity = ConvOperator()  # Identity Operator
>
> # Combine operators with coefficients
> alpha, beta = 1.0, 0.5
> D = ConvOperator()
> D.kernel = D_t.kernel - alpha * D_xx_yy.kernel - beta * D_identity.kernel
>
> # Estimate PRE from model predictions
> PRE = D(model(X))
> ```
>
> ## Advanced Experiments
>
> For Navier-Stokes, MHD, and other experiments:
> 1. Generate data: Run scripts in `Neural_PDE/Numerical_Solvers/`
> 2. Train models: Use scripts in `Neural_PDE/Expts/`
> 3. Run uncertainty estimation: See scripts in `Marginal/` and `Joint/`
>
> ## Repository Structure
>
> - `Joint/`, `Marginal/`: Conformal prediction implementations
> - `Neural_PDE/`: Neural PDE solver implementations
> - `Utils/`: Utility functions
> - `Other_UQ/`: Bayesian Deep Learning benchmarks
>
>
> ***
> ***
> # Discussion on discretisation:
>
> We've addressed the impact of spatial/temporal resolution on residual estimates in Appendix D.1, but agree this merits further discussion. The discretisation in PRE-CP stems from the neural-PDE solver itself. With pre-trained models, we typically have limited control over this aspect. The convolutional kernels (and corresponding finite difference stencils) adopt the discretisation present in the predicted data. As shown in Figure 10 for the 1D Advection equation, coarser discretisation leads to inflated error bounds compared to finer discretisation implementations. However, PRE-CP consistently delivers guaranteed coverage regardless of resolution.
> Even with coarser discretisation, PRE-CP remains valuable by:
>
> 1.	Providing statistical identification of poorer fit regions (marginal formulation)
>
> 2.	Highlighting physically inconsistent predictions (joint formulation)
>
> 3.	Enabling relative assessment of physical inconsistencies across predictions
>
> While residuals may be inflated with coarser discretisation, the corresponding bounds reflect this inflation, preserving the relative information about physical inconsistency across a series of predictions. We're currently leveraging this property to develop an active learning pipeline for neural PDEs, using PRE-CP formulation as an acquisition function.
>
> ***
> ***
> # Unknown PDE terms:
>
>
> Thank you for this insightful question. Our method fundamentally relies on formulating the nonconformity score as an equality constraint through the residual formulation, as demonstrated in the Theorem in Appendix A. This approach allows us to perform conformal prediction without requiring data.
>
> When certain PDE terms are unknown, the equality constraint may not be fully satisfied, potentially limiting PRE-CP's direct applicability. However, PRE-CP doesn't necessarily require complete knowledge of the entire PDE family. As shown in our Navier-Stokes and MHD equation examples, we can still derive meaningful bounds using only one conservation law (continuity or momentum) without explicitly incorporating all equations in the family.
>
> It's worth noting that our work primarily targets neural PDE surrogates for forward modelling where the PDE terms are known. This focus enables us to provide guaranteed uncertainty quantification for these specific applications.
>
> ***
> ***
> # Computational Overheads:
>
> The computational overhead for PDE residual evaluation is relatively low due to our use of highly optimised convolutional operations. The cost hierarchy in our workflow is:
>
> 1.	Running full simulations (e.g., 350 core hours for Tokamak plasma modelling)
>
> 2.	Training neural PDE models (e.g., 6 hours on a single A100 for FNO)
>
> 3.	Model inference (e.g., 90 seconds on a standard laptop)
>
> 4.	Residual estimation (less than 5 seconds)
>
>
> We've optimised our ConvOps library to minimise computational costs through:
>
>
> 1.	Additive kernels for linear PDE components, reducing the number of required convolutions (detailed in Appendix D)
>
> 2.	Support for both spectral and direct convolutions, providing speedups for high-resolution grids [1]
>
>
> These optimisations maintain the physics residual equality constraint and prediction exchangeability, preserving our coverage guarantees.
>
> [1] Rippel, O., Snoek, J., & Adams, R. P. (2015). Spectral Representations for Convolutional Neural Networks. arXiv preprint arXiv:1506.03767. Retrieved from https://arxiv.org/abs/1506.03767

---

### Official Review · Reviewer_Zd7y · 2025-03-17

**Overall Recommendation:** 1

**Summary:**

This paper proposes a model-agnostic, physics-informed conformal prediction network that provides guaranteed uncertainty estimates independent of input data.

**Claims And Evidence:**

1. The proposed approach is model-agnostic and physics-informed. The physics-informed aspect is evidenced in Section 4, but whether the approach is truly model-agnostic is not clearly demonstrated.
2. The proposed approach guarantees coverage bounds both marginally and jointly. This is evidenced by Theorem A.1.

**Essential References Not Discussed:**

N/A

**Experimental Designs Or Analyses:**

The authors evaluate the proposed approach using:
(1) Wave equations,
(2) Navier-Stokes equation,
(3) Magnetohydrodynamics,
(4) Plasma modeling within a tokamak,
(5) Magnetic equilibrium in a tokamak.

The estimated uncertainty of the model is compared with the PDE residual.

**Methods And Evaluation Criteria:**

The authors utilize physicist residual error as the nonconformity score, enabling data-free prediction. The proposed approach is validated through comparisons of estimated uncertainty with PDE residuals.

**Other Comments Or Suggestions:**

- Please highlight the best-performing approach in the quantitative tables in the supplementary material.
- It may be beneficial to move these tables to the main paper, as quantitative results are important for evaluating the proposed approach.

**Other Strengths And Weaknesses:**

Strengths:
- The paper introduces an interesting framework that extends conformal prediction to physics-informed models.

Weaknesses:
1. From the quantitative evaluation in the supplementary material, the proposed approach does not achieve the best performance in most scenarios compared to other baselines. This raises concerns about its effectiveness.
2. The writing of the paper needs improvement, as many symbols in Section 5 are not clearly defined.

**Questions For Authors:**

1. Uncertainty can be broadly categorized into aleatoric and epistemic. Given that the proposed approach estimates uncertainty independent of input, how does it differ from epistemic uncertainty estimation approaches?
2. Why are validation plots and quantitative analysis provided only for the first three experiments but not for plasma modeling and magnetic equilibrium?

**Relation To Broader Scientific Literature:**

The study contributes to uncertainty quantification (UQ) in physics-informed machine learning and conformal prediction. The proposed method aligns with ongoing research in Physics-informed neural networks (PINNs), neural PDE solvers, and uncertainty quantification in computational physics.

**Theoretical Claims:**

The authors claim that the proposed approach guarantees coverage bounds both marginally and jointly, as formalized in Theorem A.1.

---

> ### Author Rebuttal · Authors · 2025-03-31
>
> # Quantitative evaluation to baselines
>
> Thank you for raising this important point. We'd like to clarify that our framework indeed demonstrates superior performance in guaranteed coverage compared to baseline methods. In Appendix C (Tables 3-5), we comprehensively compare our method (CP-PRE) against standard Bayesian approaches (BNN, MC Dropout, Deep Ensembles, SWA-G) and data-driven inductive conformal prediction using absolute error residual (CP-AER).
>
> Our results show that CP-PRE consistently provides guaranteed coverage across all experiments, including both in-distribution and out-of-distribution evaluations, with performance comparable to CP-AER. While CP-AER achieves similar coverage, it requires extensive simulation data for calibration—a significant limitation.
> Following Reviewer X8Gm's suggestion, we've updated our evaluation metrics to include data generation time for CP-AER and residual estimation time for CP-PRE in the revised manuscript, providing a more complete comparison of computational requirements and is shown in the rebuttal to X8Gm.
>
> We appreciate your feedback and have moved Tables 2-5 to the main text, highlighting methods that achieve estimated coverage for improved clarity. Thank you for helping us enhance the paper's readability.
>
> ***
> ***
>
> # Missing symbols and improving writing
>
> Thank you for your careful review of Section 5. We've thoroughly re-examined this section and have ensured all symbols are properly defined. While Reviewer X8Gm found the writing quality satisfactory, we recognise the importance of clarity for all readers. In our revised manuscript, we've added additional clarification where needed and performed a comprehensive review of all notations to ensure consistency and accessibility throughout the paper. We appreciate your feedback, as it has helped us improve the overall quality of our presentation.
>
>
>
> ***
> ***
>
> # Aleatoric vs Epistemic Uncertainty
>
> Thank you for raising this important conceptual question. The uncertainty quantified by PRE-CP aligns with conformal prediction's characterization of predictive uncertainty.
> From one perspective, this could be viewed as aleatoric uncertainty since we construct confidence intervals relative to a specific probability distribution (the distribution from which calibration data, i.e. initial conditions are sampled). Alternatively, it could be considered epistemic uncertainty since we model the neural network's error through confidence intervals (typically used for unknown but fixed quantities).
> While we believe the latter interpretation is more appropriate, we acknowledge that the traditional aleatoric/epistemic dichotomy may not be directly applicable to our framework. This distinction is most valuable when both uncertainty types coexist and require separate treatment [1]. Although we could elaborate on this in our manuscript, we felt it somewhat peripheral to our paper's focus on practical uncertainty quantification rather than fundamental statistical theory.
>
>
> [1] S. Ferson, and L. Ginzburg, "Different methods are needed to propagate ignorance and variability", Reliability Engineering & System Safety, Elsevier, 1996, https://doi.org/10.1016/S0951-8320(96)00071-3
>
>
> ***
> ***
>
>
> # Validation plots for all experiments
>
> Thank you for this question. We've included complete validation plots and quantitative analysis for all experiments, including plasma modeling and magnetic equilibrium, in the appendices due to space constraints in the main text:
>
>
> •	Plasma modeling coverage plots appear in Figure 29, Appendix M
>
> •	Magnetic equilibrium coverage plots are shown in Figure 34, Appendix N
>
>
> We initially focused on presenting coverage guarantees for widely recognized test cases (Wave, Navier-Stokes, and MHD equations) where we do ablation studies in the main text. Based on your feedback, we will improve cross-referencing to these appendices to ensure readers can easily locate the complete analysis for all experiments.

---

### Official Review · Reviewer_MQ8g · 2025-03-19

**Overall Recommendation:** 3

**Summary:**

Papers consider uncertainty quantification of PDEs. They claim that by utilising a physics based approach they can quantify and calibrate the model’s inconsistencies with the PDE.

**Claims And Evidence:**

(see Other Strengths And Weaknesses))

**Essential References Not Discussed:**

.

**Experimental Designs Or Analyses:**

(see Other Strengths And Weaknesses))

**Methods And Evaluation Criteria:**

(see Other Strengths And Weaknesses))

**Other Comments Or Suggestions:**

Overall approach or "big picture" could be explained better. Perhaps showing an example which shows that how CP is calculated and then how it is used to solve practical problem? Some kind of algorithmic description of the overall approach could also be helpful.

**Other Strengths And Weaknesses:**

They claim that by utilising a physics based approach they can quantify and calibrate the model’s inconsistencies with the PDE. I can follow the paper in high level, meaning (based on my understanding) they calculate score function PRE using the stencil approximation and calculate some kind of conformal prediction (CP). But I have to admit that I do not understand the pig picture meaning that why they are doing that or how these CPs can be used in practice?

Maybe the reason is that I am not previously familiar with CP and cannot completely follow the idea based their description. Probable due to size constraints this CP section is quite short and mostly summarises some key definitions and results from previous works, but it does not explain so much that how the quantities are actually calculated or how those are used. They could explain more that how the approach is actually used.

Also, results shows PRE (which is basically just the derivate of the solution) and derived CPs for different PDE problems. Those are nice looking pictures, but again overall meaning or how those could help in practice is not clear to me. Furthermore, they also do not compare to any of previous methods.

**Questions For Authors:**

No further questions

**Relation To Broader Scientific Literature:**

PDEs are important in physics.

**Theoretical Claims:**

(see Other Strengths And Weaknesses))

---

> ### Author Rebuttal · Authors · 2025-03-31
>
> # The big picture:
>
> We appreciate this opportunity to clarify our motivation. Our work stems from the need to make neural PDE solvers more practical for scientific modelling. Numerical PDE solvers have been essential to scientific modelling since the 1950s, enabling cost-effective simulation of complex scenarios. However, they still impose significant computational burdens in complex settings.
>
> Neural PDEs offer a promising middle ground—networks that learn physics approximations could quickly explore design spaces before running expensive numerical simulations. For this workflow to be practical (as shown in Figure 1), we must quantify neural PDE model performance with reliable error bounds.
>
> While various Bayesian methods can be applied to neural PDE solvers, they fail to guarantee coverage in high-dimensional settings. Extensions of inductive conformal prediction to spatio-temporal domains require substantial calibration data for error bounds.
>
> Our approach addresses these limitations by using PDE residuals as nonconformity scores, providing guaranteed bounds across conservative PDE variables without requiring data or sampling, functioning post-hoc without modifications. Table 2 in Appendix C qualitatively compares our method's advantages against Bayesian UQ and standard CP methods.
>
> ***
> ***
> # Utility of this method and the meaning of these figures:
>
> Our approach begins with predictions of spatio-temporal physical field variables from a neural PDE solver. We then transform these predictions to their conservative form using the differential operators that characterise the PDE. This transformation captures how well the solution satisfies conservation equations, identifying regions of physical inconsistency. The Physical Residual Error (PRE) estimates reveal regions where the predicted dynamics deviate from the ground truth. By applying Conformal Prediction (CP) to these PRE values, we obtain calibrated bounds across conservative variables that provide meaningful information to domain experts. The figures illustrate how conservative variables are modelled across space and time and the regions where the model struggles to understand and learn the known physics.
>
> Additionally, our PRE-CP framework serves as an indicator of model quality, as demonstrated in Appendix I. While PRE-CP guarantees coverage regardless of model fit, the width of the error bars effectively measures model quality. Figure 18 compares (a) predictions from poorly-fit (bad) and well-fit (good) models, showing that (b) error bounds for the well-fit model are substantially narrower than (c) those of the poorly-fit model.
>
> ***
> ***
>
> # Procedure:
>
> We thank the reviewer for this helpful suggestion. Due to page constraints, we limited our discussion of conformal prediction in the main text while referencing relevant literature for broader context. We have outlined an abridged algorithmic procedure below and have added a detailed one to the appendix.
>
> 1.	Neural PDE Solver Setup: Define PDE simulator, generate data and train neural approximator
> 2.	Physics Residual Error: Calculate $|D(\hat{f}(X))|$ using differential operators as convolutions.
> 3.	Calibration:
>
>   a. Sample ${X_1,...,X_n}$, get predictions ${\hat{f}(X_1),...,\hat{f}(X_n)}$
>
>   b.	Compute PRE scores ${|D(\hat{f}(X_1))|,...,|D(\hat{f}(X_n))|}$
>
>   c.	For confidence $1-\alpha$, find $\hat{q}^\alpha$ as the $\lceil(n+1)(1-\alpha)\rceil/n$ quantile from the PRE scores.
> 4.	Application:
>
>   a. For new $X_{n+1}$, calculate $|D(\hat{f}(X_{n+1}))|$
>
>   b.	Valid if within $\mathbb{C}^\alpha = [-\hat{q}^\alpha, \hat{q}^\alpha]$
>
> 5.	Interpretation: Apply cell-wise for local bounds or use supremum for global bounds
>
> We appreciate your feedback and thank you for making the paper accessible to a broader audience.
> ***
> ***
>
> # Comparison to other methods:
>
> Our method's coverage performance is documented in Tables 3-5 in Appendix C, where we compare against both Bayesian UQ approaches (deep ensembles, MC dropout, stochastic-weighted averaging, Bayesian neural networks) and data-driven conformal prediction using absolute error residuals (CP-AER).
>
> Our method (CP-PRE) consistently provides guaranteed coverage across all experiments in both in-distribution and out-of-distribution evaluations, with performance comparable to CP-AER. However, it's important to note that CP-AER requires extensive simulation data for calibration, while our method does not. This difference is quantified in the tables, showing that the computational cost of evaluating PRE is only a fraction of the time needed to acquire new simulation data, but has been explicitly mentioned in response to reviewer X8Gm's questions.
>
> Following reviewers' suggestions, we have moved Tables 3-5 to the main paper and hope that demonstrates the advantages of our method.
>
> We apologise for the confusion that this may have caused and welcome any further clarification requests.

---

### Decision · Program_Chairs · 2025-05-01

**Decision:**

Accept (poster)

**Comment:**

The paper introduces a physics-informed conformal prediction framework for neural PDE solvers, providing model-agnostic uncertainty quantification without labeled data. Reviewers acknowledged the method's strong theoretical foundation and empirical validation across multiple PDE systems. Concerns about implementation clarity and computational tradeoffs were noted, particularly regarding practical interpretation of results. These issues do not outweigh the work's technical contributions, but should be addressed in revision. The framework advances reliable deployment of neural PDE solvers, warranting acceptance if there is room.